# Auxiliary subunits keep AMPA receptors compact during activation and desensitization

Jelena Baranovic[1,2,3], Andrew JR Plested[1,2,3]*

[1]Institute of Biology, Cellular Biophysics, Humboldt Universität zu Berlin, Berlin, Germany; [2]Leibniz Forschungsinstitut für Molekulare Pharmakologie (FMP), Berlin, Germany; [3]NeuroCure, Charité Universitätsmedizin, Berlin, Germany

**Abstract** Signal transduction at vertebrate excitatory synapses involves the rapid activation of AMPA (α-amino-3-hydroxy-5-methyl-4-isoxazole propionate) receptors, glutamate-gated ion channels whose four subunits assemble as a dimer-of-dimers. Technical advances in cryo-electron microscopy brought a slew of full-length structures of AMPA receptors, on their own and in combination with auxiliary subunits. These structures indicate that dimers might undergo substantial lateral motions during gating, opening up the extracellular layer along the central twofold symmetry axis. We used bifunctional methanethiosulfonate cross-linkers to calibrate the conformations found in functional AMPA receptors in the presence and absence of the auxiliary subunit Stargazin. Our data indicate that extracellular layer of AMPA receptors can get trapped in stable, opened-up conformations, especially upon long exposures to glutamate. In contrast, Stargazin limits this conformational flexibility. Thus, under synaptic conditions, where brief glutamate exposures and the presence of auxiliary proteins dominate, extracellular domains of AMPA receptors likely stay compact during gating.
DOI: https://doi.org/10.7554/eLife.40548.001

*For correspondence:
plested@fmp-berlin.de

**Competing interests:** The authors declare that no competing interests exist.

## Introduction

AMPA-type glutamate receptors are found at excitatory synapses throughout the mammalian brain, where they convert glutamate release into membrane depolarization. Their fast kinetics (*Colquhoun et al., 1992*; *Geiger et al., 1995*; *Taschenberger and von Gersdorff, 2000*), as well as the physical attributes of synapses (*Xu-Friedman and Regehr, 2003*), allow them to follow glutamate transients at rates above 100 Hz. However, the structural dynamics underlying their rapid signaling are unclear.

AMPA receptors are tetrameric ligand-gated ion channels. Ignoring their intracellular regions, they consist of three layers formed from distinct domains (*Figure 1A*). The two extracellular layers assemble from amino-terminal domains and ligand-binding domains, ATDs and LBDs, respectively. These adopt local dimer pairs in resting and active receptors. These dimers pack around a central twofold symmetry axis that switches into fourfold symmetry in the transmembrane ion channel layer. Although the ATD dimers associate with high affinity ($K_D$ nM to µM *Zhao et al., 2017*), the association of ATD dimers into a tetramer as well as of LBDs into dimers and tetramers is too weak to measure. In case of the LBDs, this weak association has direct functional consequences, because the LBD intra-dimer interface ruptures upon desensitization (*Sun et al., 2002*; *Armstrong et al., 2006*; *Dürr et al., 2014*; *Twomey et al., 2017b*).

The crowded and narrow synaptic cleft is scarcely wider than the receptors are tall themselves and has narrow edges (*Zuber et al., 2005*; *Tao et al., 2018*). This observation suggests that conformational dynamics of the receptor domains and their relation to synapse dimensions has implications

**eLife digest** The nearly 100 billion neurons in our brain create a complex and intricate network that can relay information in a fraction of a second. Two neurons can communicate with each other by forming a synapse, a specialised structure where the two cells come into close contact. There, the signalling neuron releases chemicals that the receiving cell captures through dedicated receptors embedded in its membrane. For example, the AMPA receptor is a complex assemblage of different subunits that quickly transmits information by opening and closing to let ions move into the receiving cell. These receptors are some of the fastest to react to the released chemicals, allowing information to be encoded swiftly. In fact, it is increasingly clear that epilepsy and deficits in mental processes can be associated with AMPA receptors having a faulty activity.

Yet, it is still unknown how exactly these proteins work. In particular, previous studies have shown that an AMPA receptor can go through dramatic changes in its structure, with the different subunits being able to spread apart widely. However, these experiments had to be conducted when the proteins were isolated from membranes and held in a cocktail of activating or deactivating molecules for hours. It is still unclear whether the results hold when AMPA receptors sit at the membrane while assembled with their partner proteins, like they normally do in the brain.

Baranovic and Plested went on to investigate this question by using 'molecular rulers'. These tiny molecules have different lengths, and they act as yardsticks: their sticky ends can attach to specific areas in the protein, helping to measure how these regions move relative to each other when the receptors are on or off. A method called patch clamp electrophysiology was used to determine how much the normal activity of the AMPA receptors was hindered by being bound by the molecular rulers.

The results showed that AMPA receptors can undergo large structural changes but these movements require time and are much reduced by partner proteins. In the brain, AMPA receptors in synapses probably lack the freedom and opportunity to move so dramatically when neurons are communicating with each other.

Ultimately, knowing how these receptors work and move may help grasp the changes in their activity that cause connections between neurons to become defective.
DOI: https://doi.org/10.7554/eLife.40548.002

in both health and disease. For example, if the extracellular layer of AMPA receptors can undergo large conformational changes rapidly, that is on the millisecond timescale of fast excitatory transmission, this could disrupt possible interactions of the extracellular domains with the pre- and postsynaptic anchoring proteins (*Elegheert et al., 2016*). Activity-dependent anchoring might be a way to regulate synaptic strength (*Constals et al., 2015*). On the other hand, slow rearrangements could be relevant for trafficking, and in disease states.

Advances in the structural biology of ionotropic glutamate receptors (iGluRs) have produced a catalogue of static conformational snapshots. Several agonist-bound structures (*Nakagawa et al., 2005*; *Dürr et al., 2014*; *Meyerson et al., 2014*) suggest that local dimers in the ATD and LBD layer move apart from each other substantially. This movement away from the central twofold symmetry axis results in an 'open' extracellular layer, with the tops of the ATD and LBD dimers splaying wide open (desensitized structure in *Figure 1A*). The timescale of this broad lateral movement is unknown, because the structural experiments necessarily took place over hours. We therefore set out to investigate the conformational range of agonist-bound AMPA receptors with the aim of distinguishing frequently-visited, short-lived conformations from the long-lived ones that likely have less direct relevance to synaptic transmission.

Previously, we demonstrated disulphide bonds and metal bridges trapping receptors in compact, as opposed to 'open' LBD arrangements (*Salazar et al., 2017*; *Baranovic et al., 2016*; *Lau et al., 2013*). To measure the separation of domains in this work, we used bifunctional methanethiosulfonate cross-linkers (bis-MTS) of defined lengths (*Loo and Clarke, 2001*; *Guan et al., 2002*; *Armstrong et al., 2006*; *Tajima et al., 2016*) (*Figure 1D*). These cross-linkers show specific combination with two free thiol groups, provided by cysteine residues that we engineered. The selective reactivity of a series of probes can in principle report distances, giving them the property of

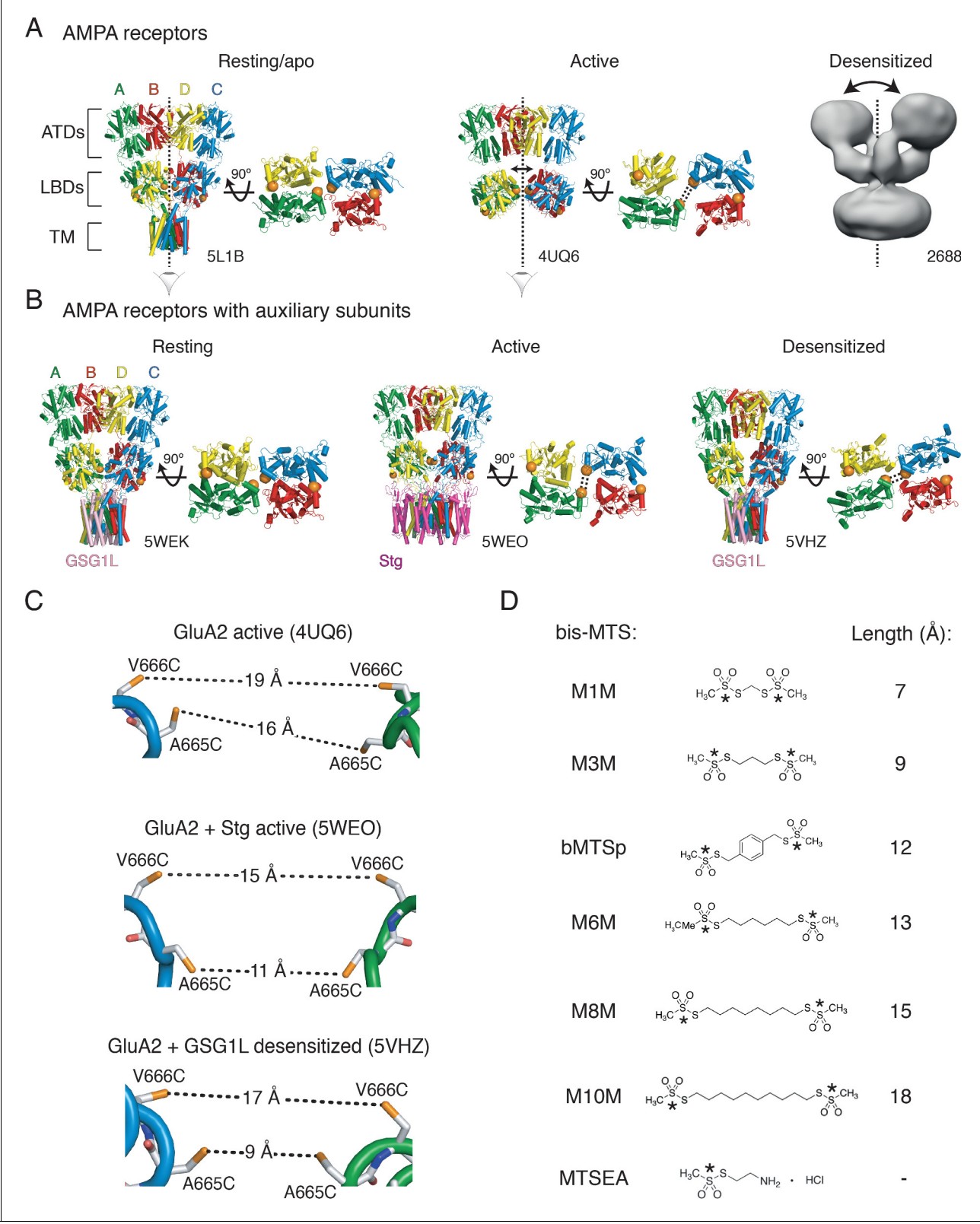

**Figure 1.** Geometry of AMPA receptors. (**A**) Structures of full-length AMPA receptors in resting, active and desensitized states. Accesion codes for PBD or EMDB are indicated. Subunits are color-coded: A – green, B – red, C – blue and D – yellow. Brackets delineate AMPA receptor domains: ATDs – amino terminal domains, LBDs – ligand binding domains and TM – transmembrane region. The cytoplasmic domain is not resolved. Vertical, dashed lines indicate central twofold symmetry axis. The corresponding LBDs are shown in the bottom-up view. Double-headed arrows indicate movements of

*Figure 1 continued on next page*

*Figure 1 continued*

dimers in the extracellualr layer. Each orange sphere indicates position of residues 665 and 666 (mutated individually in this study). In subunits A and C, the mutated residues are facing each other across the LBD inter-dimer interface, whereas in subunits B and D, they are located on the outer surface of the protein. Based on the structures, the mutated residues could cross-link across subunits A and C, as symbolized by the double dashed lines for the functional states studied here. The measured distances are shown in (**C**). Details are omitted for the desensitized structure (2688) due to its low resolution. (**B**) Same as in (**A**) but for GluA2 structures complexed with auxiliary subunits: Stg – Stargazin (dark purple) and GSG1L (light purple). (**C**) Distances between sulfhydryl groups (orange) of individually mutated residues, A665C and V666C, in agonist-bound strucures shown in panels A and B. (**D**) Structures of bifunctional (bis-MTS: M1M-M10M) and monofunctional (MTSEA) compounds. Lengths were measured between reactive sulphur atoms (SG, asterisks).

DOI: https://doi.org/10.7554/eLife.40548.003

nanometre-scale molecular rulers. Based on activated and desensitized state structures, we reasoned that, crosslinking between ligand binding domain dimers should be state-dependent. Short cross-linkers should be favored in active states with compact LBD layers, and longer crosslinkers should be favored in desensitized states.

Our results suggest that the more opened-up a given conformation of the extracellular layer of the receptor is, the slower it is to access. Once attained, these conformations are stable. However, auxiliary subunits restrict the conformational ensemble, maintaining more compact arrangements of the LBD layer, without much separation of the two LBD dimers. This kinetic classification suggests AMPA receptors at synapses probably have similar, compact geometries regardless of their instantaneous gating state.

## Results

### The LBD layer opens up in desensitized AMPA receptors

The rupture of the LBD intra-dimer interface is a structural hallmark of AMPA receptor desensitization, as shown by biophysical studies based on the structures of isolated ligand binding domains (*Sun et al., 2002*; *Armstrong et al., 2006*). Several cryo-electron microscopy (cryo-EM) structures of full-length receptors (*Nakagawa et al., 2005*; *Dürr et al., 2014*; *Meyerson et al., 2014*), such as the one shown in *Figure 1A*, suggest that desensitization might involve further rearrangements of the ligand-binding domains, including movement of the two LBD dimers away from the central two-fold axis of symmetry and an even more substantial 'opening' of the ATD-layer parallel to the membrane plane.

We attempted to capture this movement between LBDs with bis-MTS cross-linkers ranging from 7 to 18 Å in length (*Figure 1D*). If the LBD layer is opened up in the horizontal plane upon receptor desensitization, this movement should create access for bis-MTS cross-linkers into the inter-dimer space (orange spheres in *Figure 1A*). Because bis-MTS cross-linkers are flexible (apart from bMTSp, which is rigid), they can bend and bind to Cys residues whose separation is shorter than the length of the cross-linker. Here, we assume that a fully reacted cross-linker will exert its effect mainly by preventing the engineered residues from separating more than the extended length of the cross-linker (*Figure 1D*). Conversely, if the two free Cys residues never separate beyond the length of the cross-linker during gating, then the cross-linker should have little or no effect.

The ability of the cross-linkers of different lengths to access the inter-dimer space in the LBD layer reports on the lateral separation of the two LBD dimers, but does not inform the movements the LBD layer undertook to arrive at the specific conformation. We can assume that the membrane-bound TM region somewhat restricts vertical displacement of the LBDs. Thus, the movements we are probing are those approximately parallel to the membrane. The cross-linkers could in principle be applied to the ATDs, but ATD layer is functionally silent (*Pasternack et al., 2002*) and its cross-linking does not produce measurable changes in the receptor activity (*Yelshanskaya et al., 2016*). We, therefore, do not make any extrapolations here about the ATD movements from cross-linking the LBD layer, but structural models and fluorescence studies (*Shaikh et al., 2016*) so far indicate that the ATDs do follow movements of the LBD layer (*Figure 1A–B*).

Based on the structural models of homomeric, full-length GluA2 receptors, we identified positions 665 and 666 (in the loop connecting α-helices F and G in the LBDs) as best positioned to follow

separation of the LBD dimers (*Figure 1A–C*). The presumptive geometries of the sulfhydryl groups (SG) for cysteine mutants at these sites, in the agonist bound states of GluA2 receptor (with and without auxiliary subunits) are shown in *Figure 1C*. The structure of the desensitized GluA2 receptor (EMDB: 2688, *Figure 1A*) is not detailed enough to measure residue distances, but the equivalent residues are 21 Å apart in homologous kainate receptors (PDB: 5KUF; *Meyerson et al., 2016*). Based on these structural data, all six bis-MTS cross-linkers should be able to modulate desensitized V666C and A665C receptors by preventing engineered Cys at both sites from reaching their full separation in the desensitized state, in the absence of auxiliary subunits. Given the distances in published structures, one might expect M10M to stabilize active V666C receptors in the absence of auxiliary subunit Stargazin and M8M in its presence (*Figure 1C–D*).

As shown in *Figure 2B and D*, bis-MTS cross-linkers M1M to M10M all caused strong reduction of the peak current in V666C mutant when applied in the desensitized state. To quantify this effect, the peak current was measured from the control pulses before ($I_{peak}$ pre-trap, four pulses) and after the 1 min application (trap) of the cross-linker ($I_{peak}$ post-trap, second control pulse after the trap, arrows in *Figure 2A–C*). For each patch, the ratio of $I_{peak}$ post-trap over $I_{peak}$ pre-trap was determined and plotted as shown in *Figure 2D–E*.

For the V666C mutant, bis-MTS cross-linkers from 7 to 15 Å in length (M1M-M8M; 1 µM), inhibited ~90% of the peak current in the patch after a 1-min application (*Figure 2—source data 1*). The longest (M10M) cross-linker was the slowest one to act (*Figure 2F*), leading to less inhibition (~70%) in the first minute of exposure. The reduction was less pronounced for A665C mutant for all cross-linkers (~50%, *Figure 2E* and *Figure 2—source data 1*).

The slow reaction time of bis-MTS cross-linkers ($\geq$minute, *Figure 2F*) is a direct consequence of their low concentration (1 µM). We chose a low concentration in order to minimize the linking of these highly reactive compounds to each other (*Kenyon and Bruice, 1977*). However, the slow action of bis-MTS cross-linkers could also be due to the sampling of slowly attained conformations. To determine that modification of the V666C receptors by MTS cross-linkers can proceed on the same time scale as receptor gating, we performed additional trapping experiments with 50 µM M3M and M10M (*Figure 3*). If the slow reaction rate of bis-MTS is primarily due to their low concentration, the reaction time should increase accordingly at higher concentrations. Indeed, at 50 µM, both bis-MTS cross-linkers were roughly 50x faster to modify the receptors ($\tau_{M3M}$ = 0.2 s and $\tau_{M10M}$ = 1.7 s, *Figure 3B and D*). The relative modification time was preserved with the longer cross-linker (M10M) still being slower than the shorter one (M3M). These experiments indicate that bifunctional MTS reagents are capturing rapid transitions in receptor structure on the millisecond timescale.

Inhibition was overall so profound that we sought to establish that it was specific. Two other factors potentially contribute to the current decrease: non-specific run-down of the current and disulphide bonding of the introduced cysteines to each other. Current run-down is particularly difficult to avoid in the long records that we made for these experiments. A665C and V666C sulfhydryl groups were both previously shown to crosslink in the presence of oxidizing agent CuPhen (*Salazar et al., 2017*; *Lau et al., 2013*; *Yelshanskaya et al., 2016*). To account for these confounding factors, we made paired recordings: a patch was first exposed to a 1-min long trapping pulse containing glutamate only (no cross-linkers), followed by trapping of the same patch in glutamate and a cross-linker (*Figure 2B–C*). Any run-down in the patch or possible cross-linking of the cysteines to each other was then assessed from trapping in glutamate only. In case of A665C, the introduced cysteines linked to each other during long exposures to 10 mM glutamate, resulting in the peak current reduction of $0.62 \pm 0.04$ ($n = 23$, $p < 10^{-7}$ vs. WT: $0.97 \pm 0.03$, $n = 30$), consistent with previous results (*Lau et al., 2013*; *Yelshanskaya et al., 2016*). In reducing conditions, the trapped A665C receptors recovered within seconds (grey dots in *Figure 2C*, $\tau$ = $3.3 \pm 0.4$ s, $n = 17$). V666C mutant underwent less peak current inhibition during exposures to 10 mM glutamate (V666C: $0.86 \pm 0.02$, $n = 44$, $p = 0.003$ vs. WT) and showed no evidence of recovery in reducing conditions. For V666C mutant, all six bis-MTS cross-linkers resulted in more current inhibition than glutamate alone (p $\leq$ 0.03 with and without bis-MTS, depending on the cross-linker; paired randomization test). Overall, paired recordings gave indistinguishable results to the non-paired recordings. We therefore pooled the cross-linking data in glutamate alone across conditions for each mutant (*Figure 2D–E* and *Figure 2—source data 1*).

To confirm that the observed peak current reduction came from crosslinking rather than monofunctional engagement of a cross-linker, we modified mutants with MTSEA (*Figure 1D*), which can

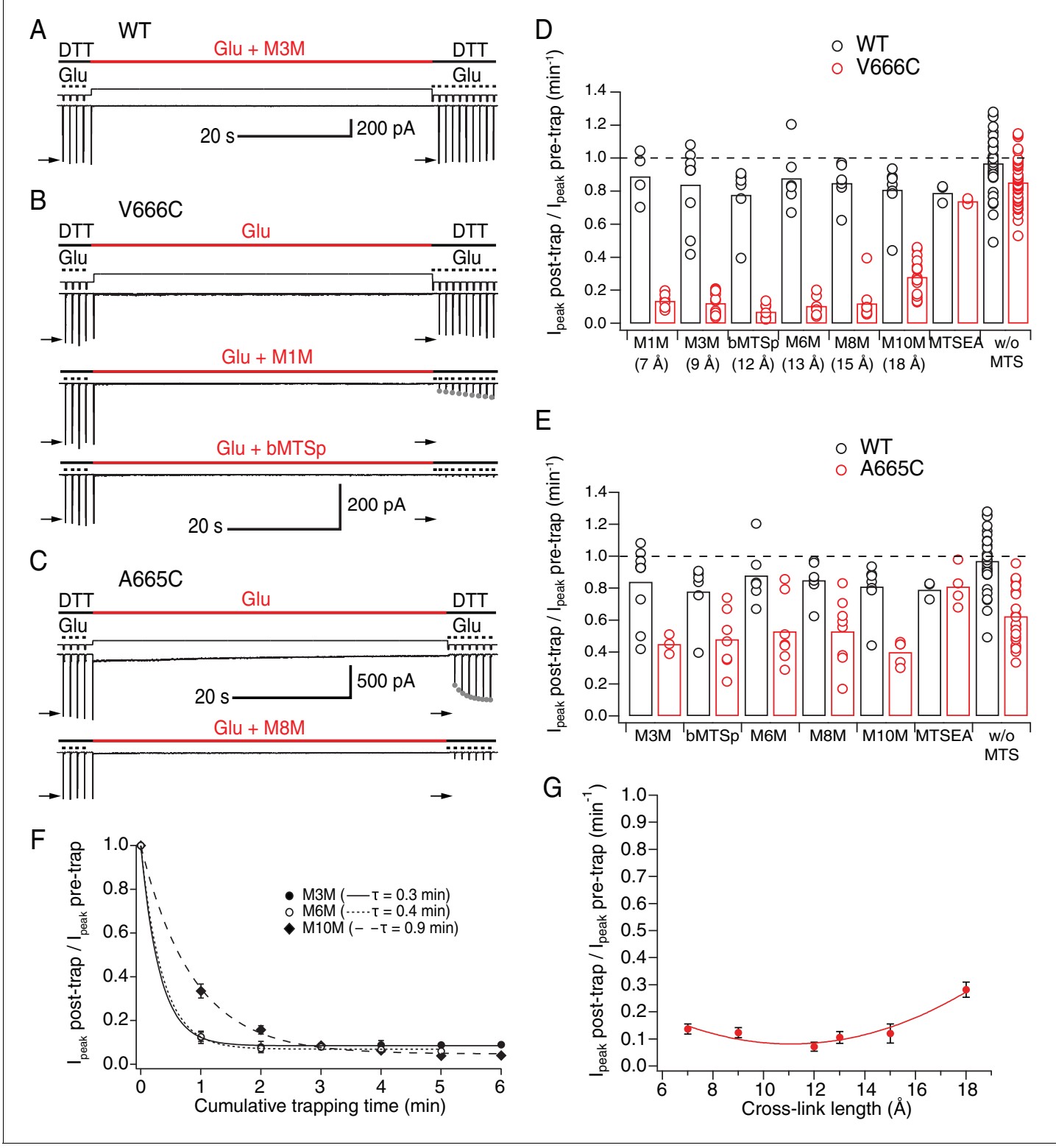

**Figure 2.** LBDs can separate ≥18 Å in desensitized GluA2. (**A**) Control recording of wild-type (WT) GluA2 receptors in response to a trapping protocol. Movements of the piezo reflecting the solution exchange are shown in thin, black lines above the current trace. Composition of the solutions is indicated in thick lines above the piezo trace. Downward ticks are 200 ms control jumps from DTT (1 mM) to DTT and glutamate (Glu, 10 mM). Four pre-trap control pulses were followed by a 1 min long trapping pulse (red line) in Glu (10 mM) and M3M (1 μM). After the trapping pulse, the patch was exposed to 10 post-trap control pulses. The first post-trap pulse gives no response because receptors are desensitized (for details, see Materials and

*Figure 2 continued on next page*

*Figure 2 continued*

methods). (B) Same as in (A), but for V666C mutant. The top two recordings are from the same patch: in the first trace, V666C receptors were exposed only to Glu. Trapping of the same patch in Glu and M1M (1 μM), results in pronounced peak current reduction, which partly recovered with $\tau_{recovery}$ = 30 ± 7 s, n = 5 (grey dots; post-trap control pulses extended to 30 for fit). The bottom trace is a different patch trapped in bMTSp (1 μM), showing even stronger peak current reduction without any recovery. (C) As in (B) but for the A665C mutant. The two traces are paired recordings of the same patch. Post-trap control pulses show that A665C cross-links to itself, but most of the current recovers within several seconds after the trap (grey dots; $\tau_{recovery}$ = 3.3 ± 0.4 s, n = 17). If the same patch is now trapped in Glu and M8M (1 μM), the peak current reduction is much more pronounced and does not recover. (D) Summary of the trapping effects for WT (black) and V666C (red) for cross-linkers M1M (7 Å) to M10M (18 Å). Trapping effect after 1 min was calculated as the ratio of the post-trap and pre-trap peak current (arrows in A-C). MTSEA is a monofunctional reagent and 'w/o MTS' stands for 'without MTS' (Glu only, pooled for all experiments). Dashed line indicates no effect. For peak current reduction in a bis-MTS vs. w/o MTS (pooled), $p < 10^{-7}$, for all cross-linkers. (E) Same as in (D), but for A665C (red) in cross-linkers M3M (9 Å) to M10M (18 Å). A665C mutant in the presence and absence of an MTS cross-linker resulted in $p \leq 0.02$, depending on the cross-linker. For statistics vs. WT and between different bis-MTS, see *Figure 2—source data 1*. (F) Trapping time for V666C receptors in M3M, M6M and M10M. The 1-min trapping protocol was repeated up to six times, resulting in a cumulative exposure of the patch to a bis-MTS of up to 6 min. The data were fit with a monoexponential for each cross-linker (τ indicated in brackets). (G) Trapping profile of desensitized V666C receptors shows the effect of each cross-linker vs. its length, in the first minute of exposure. The data were fit with a parabola (red line): $f(x) = K_0 + K_1*(x - K_2)^2$ (for details, see Materials and methods).
DOI: https://doi.org/10.7554/eLife.40548.004

The following source data and figure supplements are available for figure 2:

**Source data 1.** Statistics of trapping desensitized wild-type (WT), A665C and V666C receptors with different bis-MTS cross-linkers; data from *Figure 2D–E*.
DOI: https://doi.org/10.7554/eLife.40548.008

**Figure supplement 1.** Effects of bis-MTS cross-linkers on S662C and I664C mutants.
DOI: https://doi.org/10.7554/eLife.40548.005

**Figure supplement 2.** Bis-MTS cross-linkers potentiate K493C currents by accessing intra-dimer space.
DOI: https://doi.org/10.7554/eLife.40548.006

**Figure supplement 3.** Bis-MTS compounds do not form inter-receptor cross-links.
DOI: https://doi.org/10.7554/eLife.40548.007

interact only with a single cysteine. As shown in *Figure 2D–E*, MTSEA failed to inhibit either V666C or A665C above Glu-only control (V666C: 0.74 ± 0.008, n = 4, p = 0.1; A665C: 0.81 ± 0.06, n = 4, p = 0.06). Thus, bifunctional MTS cross-linking was necessary for the peak current reduction observed in desensitizing AMPA receptors.

The effect of all bis-MTS cross-linkers with respect to their length is summarized in the trapping profile for desensitized V666C receptors (*Figure 2G*). The profile is a shallow parabola with strong effects across all tested lengths. Fitting a parabola to these data is justified by the observation that direct disulphide crosslinking (representing the short distance limit) is quite ineffective at this site (*Figure 2D* and *Lau et al., 2013*). These results demonstrate that limiting V666C separation of desensitized AMPA receptors to 18 Å causes inhibition, indicating further separation of the residues is possible, in good agreement with cryo-EM structures (*Meyerson et al., 2014*; *Dürr et al., 2014*).

In addition to positions 665 and 666, we also tested nearby positions 662 and 664 for their sensitivity to bis-MTS cross-linkers (*Figure 2—figure supplement 1*). The I664C mutant showed similar levels of the peak current reduction to V666C. However, structural models position V666C sulfhydryls at distances that are better matched by the length of bis-MTS cross-linkers (*Figure 1C–D*) than the separation of I664C sulfhydryls across different functional states (*Figure 2—figure supplement 1A*). Like the A665C mutant, the S662C mutant showed considerable disulphide formation. For these reasons, we focused on the V666C mutant in further investigations.

Next, we tested if the cross-linkers specifically target introduced cysteines. We utilized the established phenomenon that restraining the LBD intra-dimer interface leads to block of desensitization (*Sun et al., 2002*; *Armstrong et al., 2006*). If the bis-MTS cross-linkers specifically target intra-dimer interface, then introducing a single cysteine mutant, positioned within LBD dimers (*Figure 2—figure supplement 2A*), such as K493C (*Armstrong et al., 2006*), should lead to a block of desensitization depending on the length of the cross-linker. Sulfhydryl groups of K493C residues in resting and active GluA2 receptors with intra-dimer LBD interface intact are 7 and 6 Å apart, respectively (as measured in the following resting: 3KG2, 4U2P, 5L1B and active structures: 4UQ6, 5WEO, using the most common rotamers). The desensitized structure in complex with the auxiliary subunit GSG1L has K493C residues 12 Å apart (5VHZ). It has been previously shown that sulfhydryl groups of residues

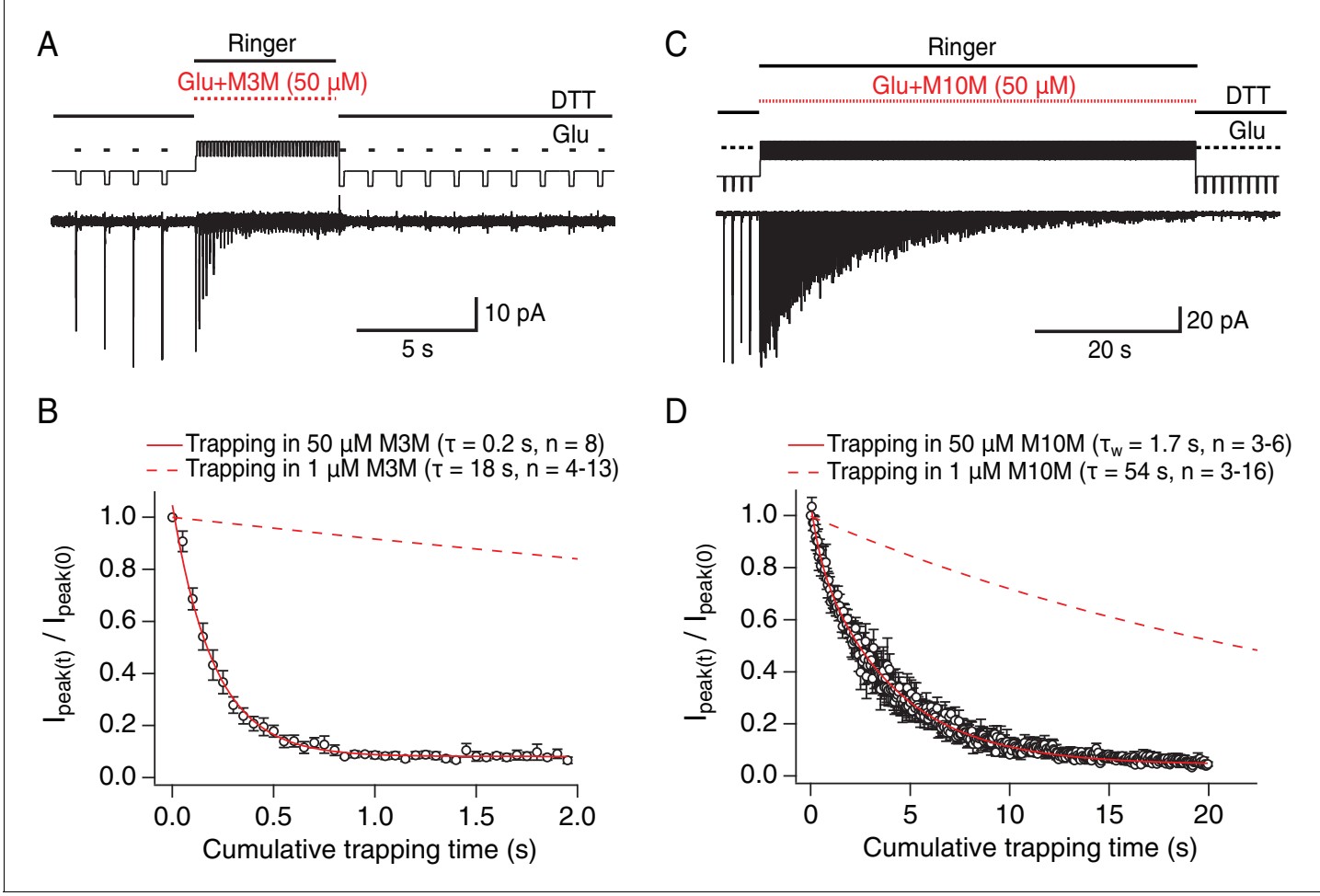

**Figure 3.** Bis-MTS cross-linkers trap in milliseconds. (**A**) Trapping protocol for V666C receptors in 50 µM M3M and 10 mM glutamate (Glu), same as in *Figure 2A–C*, but with a trapping jump consisting of 50 ms-pulses into 50 µM M3M and 10 mM Glu at 6.7 Hz (thick, red lines), interspersed with 100 ms intervals in Ringer solution. (**B**) Progression of the peak current reduction in response to the trapping pulses (red in (**A**)) was determined for each patch, normalized to the first pulse, averaged across patches and plotted. The resulting decrease in the peak current was fit with a monoexponential (red line) with $\tau = 0.2$ s. Dashed line is a fit to trapping in 1 µM M3M from *Figure 2F*. (**C**) As in (**A**), but for M10M, with the trapping jump prolonged to 1 min to complete the peak current reduction. (**D**) As in (**B**), but for M10M. Peak current decay was best fit with a double exponential resulting in weighted $\tau_W$ of 1.7 s.

DOI: https://doi.org/10.7554/eLife.40548.009

K493C, in the absence of auxiliary subunits, must separate more than 13 Å for the LBD dimers to desensitize (*Armstrong et al., 2006*). Based on these numbers, cross-linkers that are $\leq$ 13 Å in length (M1M, M3M, bMTSp and M6M) are expected to block desensitization. When applied on K493C receptors (1 µM, for $\geq$ min), bis-MTS cross-linkers had profoundly different effect from the mutants at the inter-dimer interface: they potentiated rather than inhibited the current (*Figure 2—figure supplement 2B–D*). M1M blocked desensitization, as expected, although not completely ($I_{steady-state}/I_{peak} = 49.6 \pm 4\%$, $n = 4$, $p = 0.006$ vs. K493C desensitization levels in the absence of bis-MTS: $26.6 \pm 2\%$, $n = 50$). At 9 Å, M3M was the most effective in blocking desensitization ($I_{steady-state}/I_{peak} = 80.8 \pm 2\%$, $n = 16$, $p < 10^{-7}$ vs. bis-MTS-free desensitization). M6M cross-linker blocked desensitization, as predicted ($I_{steady-state}/I_{peak} = 63.7 \pm 7\%$, $n = 9$, $p < 10^{-7}$ vs. bis-MTS-free desensitization), but rigid bMTSp at 12 Å failed to block desensitization ($I_{steady-state}/I_{peak} = 22.3 \pm 1\%$, $n = 7$, $p = 0.5$ vs. bis-MTS-free desensitization). As expected, none of the longer cross-linkers (M8M and M10M) reduced desensitization compared to the non-treated K493C receptors (M8M: $18.3 \pm 3\%$, $n = 9$ and M10M: $40.3 \pm 6\%$, $n = 3$, $p$ 0.1 for both). Although bMTSp and M8M did not reduce desensitization, they both potentiated K493C currents (they increased steady-state and peak current

to the same extent, *Figure 2—figure supplement 2B*), unlike M10M cross-linker, which produced no change in K493C currents. The absence of the effect for M10M could be either because it is too long to bind to K493C receptors or because, even when bound, its length does not restrict movements of desensitizing LBD monomers at position 493. The bis-MTS length-dependent block of desensitization in K493C receptors strongly indicates the cross-linkers target introduced Cys mutations selectively.

We also considered the possibility that bis-MTS cross-linkers might be spuriously cross-linking to wild-type cysteines on the receptor or forming inter-receptor cross-links (*Figure 2—figure supplement 3A*). If this were the case, the peak current reduction effect would be expected to scale with the number of the receptors in the membrane, that is peak current, but no such correlation was found (*Figure 2—figure supplement 3B*). In addition, longer cross-linkers would be expected to be more efficient in forging inter-receptor cross-links, but we found that the longest cross-linker, M10M, was the slowest to react on desensitized AMPA receptors (*Figure 2F*). The absence of the strong peak current reduction in the WT receptors also speaks against the bis-MTS cross-linkers interacting with native cysteine residues. These results led us to conclude that the bis-MTS cross-linkers can cross-link cysteines introduced at the LBD inter-dimer interface (V666C).

## Cross-linked desensitized states are highly stable

Upon establishing specific and strong reduction of the peak current in desensitizing V666C receptors by the bis-MTS cross-linkers, we next sought to examine the stability of trapped states. The more stable the trapped state, the longer we would expect that it takes for trapping effects to reverse, and vice versa. The fastest recovery after trapping was observed with M1M ($\tau = 30 \pm 7$ s, $n = 5$) and the time constant of the peak current recovery could be measured by directly fitting the peak current of the post-trap control pulses (grey dots in *Figure 2B*). With longer bis-MTS cross-linkers, the recovery time increased to minutes, making direct measurements of the recovery time from post-trap control pulses impractical. Instead, the experimental design was adjusted to allow measurements of long recovery times as described in Materials and methods and *Figure 4*. After the receptors were trapped with M3M for 1 min, the peak current in the patch did recover, but very slowly, taking over 10 min (600 applications of Glu) to reach the pre-trapping levels. With longer cross-linkers, the peak current was essentially irreversible over the timescales we could measure (20 min; *Figure 4D–G*). The progressive inability of the receptors to recover from trapping with longer cross-linkers indicates increasingly stable trapped states corresponding to greater separation of the cross-linked subunits in the desensitized state (*Figure 4G–H*). Desensitized AMPA receptors with a disulphide bridge formed between the two V666C residues at inter-dimer interface fit in well with this trend, recovering in seconds following 100 s exposure to the oxidizing agent CuPhen (*Figure 4H*). Multiple stable desensitized states trapped by bis-MTS are a possible explanation for the low-resolution cryo-EM structures of desensitized AMPA receptors in the absence of auxiliary subunits (*Meyerson et al., 2014*; *Dürr et al., 2014*).

To ensure that the lack of recovery was not due to a limited reducing capacity, we tested a higher concentration of the reducing agent DTT (5 mM instead of 1 mM). Stronger reducing conditions did not consistently promote recovery of V666C receptors trapped by a long cross-linker M6M (12 Å; $p = 0.3$, *Figure 4—figure supplement 1*).

## Activation limits conformational heterogeneity of the LBD layer

We next investigated if the extracellular layer of activated V666C receptors is also accessible to a similarly wide range of bis-MTS cross-linkers. According to the cryo-EM structure of AMPA receptor in complex with glutamate and a desensitization blocker (*Meyerson et al., 2014*) (*Figure 1C*), V666C residues are 19 Å apart and therefore we reasoned that all six cross-linkers should restrain the receptor with M10M (18 Å) expected to stabilize the active conformation.

To maintain the active state, we blocked desensitization with cyclothiazide (CTZ, 100 μM). As shown in *Figure 5*, block of desensitization reduced inhibition by bis-MTS cross-linkers in the first minute of exposure (see *Figure 5* – source data 1for statistics). Current inhibition for every cross-linker in the presence and absence of desensitization is shown in *Figure 5B*. With M1M, about 15% of V666C receptors recovered from inhibition with a time constant of $\tau = 7.1 \pm 1$ s, $n = 4$, leading to the final inhibition of $0.65 \pm 0.02$, n = 12 (not shown). For M3M, the recovery was still faster with

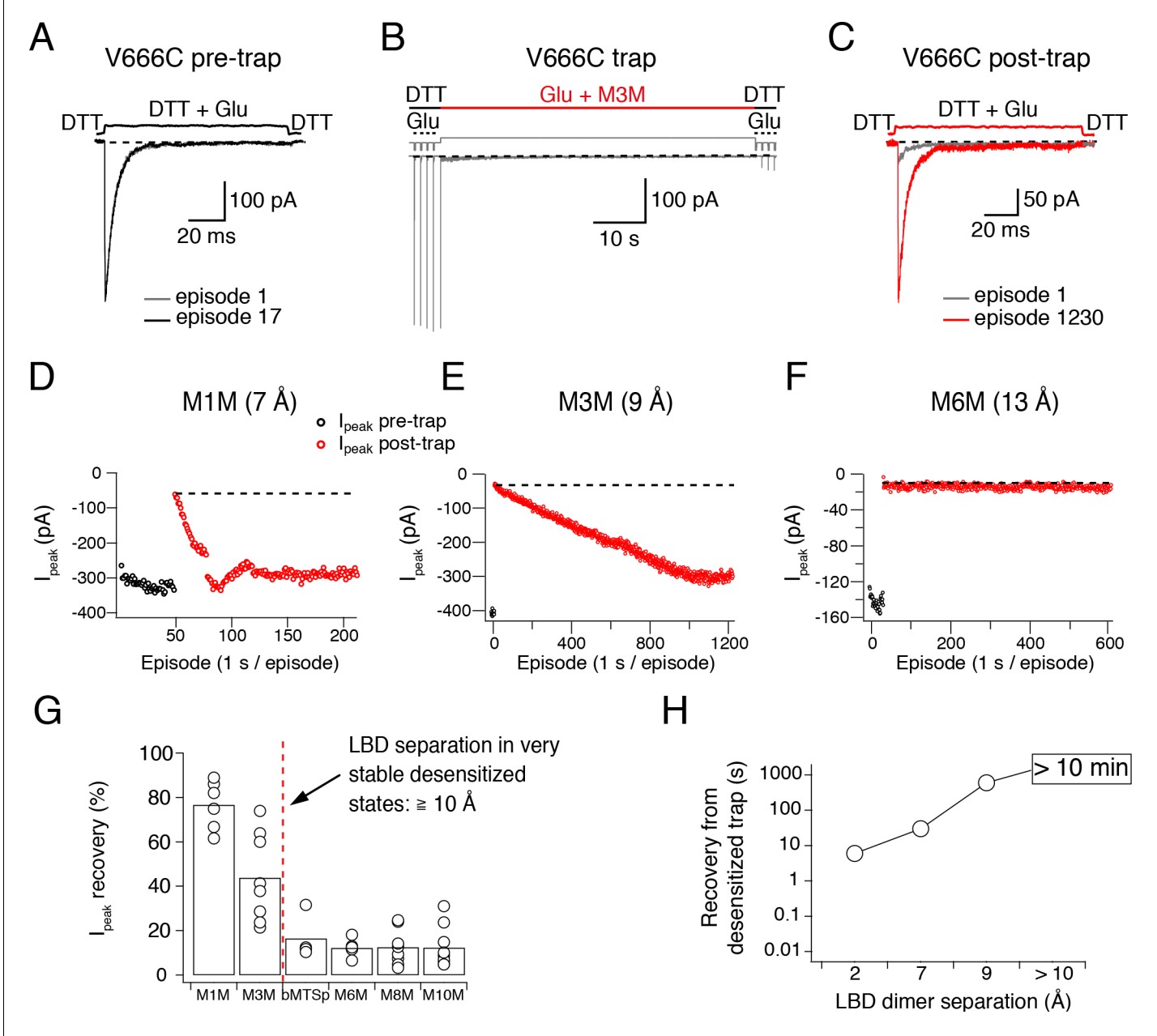

**Figure 4.** Recovery of trapped desensitized V666C receptors depends on the LBD separation. (**A**) To obtain a stable baseline response to glutamate, we first repeated brief glutamate applications in reducing conditions (1 mM DTT). In the given example, we gave 17 pulses (100 ms, 1 Hz). (**B**) In the following step, the patch was exposed to 1 μM cross-linker (here M3M) and 10 mM Glu for 1 min, with control pulses before and after the trap. (**C**) After the trapping protocol, the patch was again exposed to fast, reducing Glu jumps like in (**A**) in order to follow recovery of the response. In this example, we could record 1230 consecutive episodes (~20 min) and obtain almost complete recovery. Note the difference between the current amplitude in the 1st episode (grey) and the 1230th episode (red). The recovery of the patch current ($I_{peak}$) in typical experiments for different cross-linkers is plotted in panels D - F (panel E is the same patch as in panels A - C). Black dots show the responses before the trap and red dots the peak current after the trap. The gap in red dots in (**D**) represents a switch between recording protocols. (**G**) Summary of the peak current recovery for different cross-linkers. The percentage of recovered current is the ratio of the peak currents recorded 3–10 min after the trap to the peak current before trapping. Dashed, red line denotes a limit after which no recovery of the current could be measured within 10 min after the trap. (**H**) Plot of recovery time from trapped desensitized states (as described above and in Materials and methods) vs. the inter-dimer separation at position V666C in the LBD layer. The first data point indicates recovery from V666C disulphide bridges formed in desensitized state.

DOI: https://doi.org/10.7554/eLife.40548.010

The following figure supplement is available for figure 4:

**Figure supplement 1.** Stronger reducing conditons do not increase recovery of desensitized V666C receptors trapped by M6M.

*Figure 4 continued on next page*

*Figure 4 continued*

DOI: https://doi.org/10.7554/eLife.40548.011

$\tau = 1.98 \pm 0.2$ s, $n = 8$ (grey dots in *Figure 5A*), with about 15% the receptors recovering and leading to the final inhibition of $0.53 \pm 0.03$, $n = 9$. The fast recovery indicates that with dimer separation of about 9 Å, active receptors were trapped in an unstable, stereochemically strained state. Notably, the SG of V666C are 9 Å apart in a structure solved with the partial agonist NOW, which may represent a pre-open state (*Yelshanskaya et al., 2014*).

With desensitization blocked, AMPA receptors displayed a distinct trapping profile from that of desensitized receptors (*Figure 5C*). All cross-linkers modulated active state less than the desensitized in the first minute of exposure, apart from M8M (15 Å; *Figure 5B*) and a fraction of active receptors trapped with the shorter cross-linkers (M1M and M3M) recovered fast from the trap. The trapping profile against linker length was sharper, indicative of less conformational heterogeneity. At 12–15 Å (bMTSp, M6M and M8M), active receptors were trapped most effectively and without fast recovery; at 18 Å (M10M) the extent of trapping decreased (*Figure 5—source data 1*). This places the preferred V666C sulfhydryl separation in the active state around 15 Å, somewhat closer than 19 Å obtained in the cryo-EM structure of active GluA2 (*Figure 1C*). The description of the trapping profile by a parabola (with minimum at 13 Å) was not as good as for the desensitized condition, with the profound trapping by M8M being underestimated.

The generally reduced inhibition of active receptors, compared to desensitized ones, led us to investigate the possibility that bis-MTS reagents modify non-desensitizing V666C receptors in other ways than current amplitude reduction. We measured the rate of receptor deactivation before and after the trap in M10M in the presence of desensitization blocker CTZ and found no difference (*Figure 5—figure supplement 1C–D*, $\tau_{\text{pre-trap}} = 1.7$ ms $\pm 0.2$, $\tau_{\text{post-trap}} = 1.6 \pm 0.2$ ms, $n = 8$, $p = 0.05$ (paired randomization test)). We considered the possibility that non-desensitizing (CTZ bound) receptors were modified by M10M in a functionally silent manner. We tested this scenario with the following experiment: a patch with V666C receptors was first trapped in M10M and CTZ; CTZ was then washed-out of the patch until V666C receptors were able to desensitize again freely (*Figure 5—figure supplement 1A* (2)). After this, we exposed receptors to M10M and glutamate (*Figure 5—figure supplement 1A* (3)). If non-desensitizing V666C receptors had been silently modified by M10M, then a fraction of the receptors should have been protected resulting in the reduced sensitivity to further trapping by M10M. V666C receptors initially exposed to M10M in the presence of CTZ were modified to the same extent as naïve receptors ($I_{\text{peak}}$ post-trap/$I_{\text{peak}}$ pre-trap for V666C initially trapped in CTZ: $0.23 \pm 0.04$, $n = 6$ and for V666C never trapped in CTZ: $0.30 \pm 0.03$, $n = 16$, $p = 0.07$; *Figure 5—figure supplement 1B*). Taken together, these results strongly suggest that non-desensitizing V666C receptors were not silently modified by M10M. Instead, the reduced inhibition of active receptors reflected state-dependent protection from modification. The effect of M10M was inhibitory rather than potentiating, despite its length (18 Å) matching closely the predicted V666C separation of 19 Å (*Figure 1C*).

## Auxiliary subunits do not alter the geometry of desensitized receptors

Trapping with bis-MTS cross-linkers so far indicated more conformational flexibility of the LBD layer in desensitized than activated AMPA receptors. However, synaptic AMPA receptors are rarely expressed alone and are instead associated with various auxiliary proteins that re-define their kinetic properties (*Schwenk et al., 2012*; *Jackson and Nicoll, 2011*). We therefore wondered if the presence of auxiliary subunits, such as Stargazin (Stg), could affect conformational flexibility of AMPA receptors.

Recently, cryo-EM structures of AMPA receptors in complex with auxiliary subunits have been acquired in resting, active and desensitized states (*Figure 1B*) (*Chen et al., 2017*; *Twomey et al., 2017a*). These structures all estimate reduced separation of V666C residues when compared to receptors without auxiliary proteins. For example, sulfhydryl groups of V666C residues on subunits A and C are 19 Å apart in the activated receptors without Stg and 15 Å in complex with Stg (*Figure 1C*). In the desensitized state, the equivalent residues are 21 Å apart in homologous kainate receptors (PDB: 5KUF [*Meyerson et al., 2016*]) and 17 Å in desensitized GluA2 receptors associated

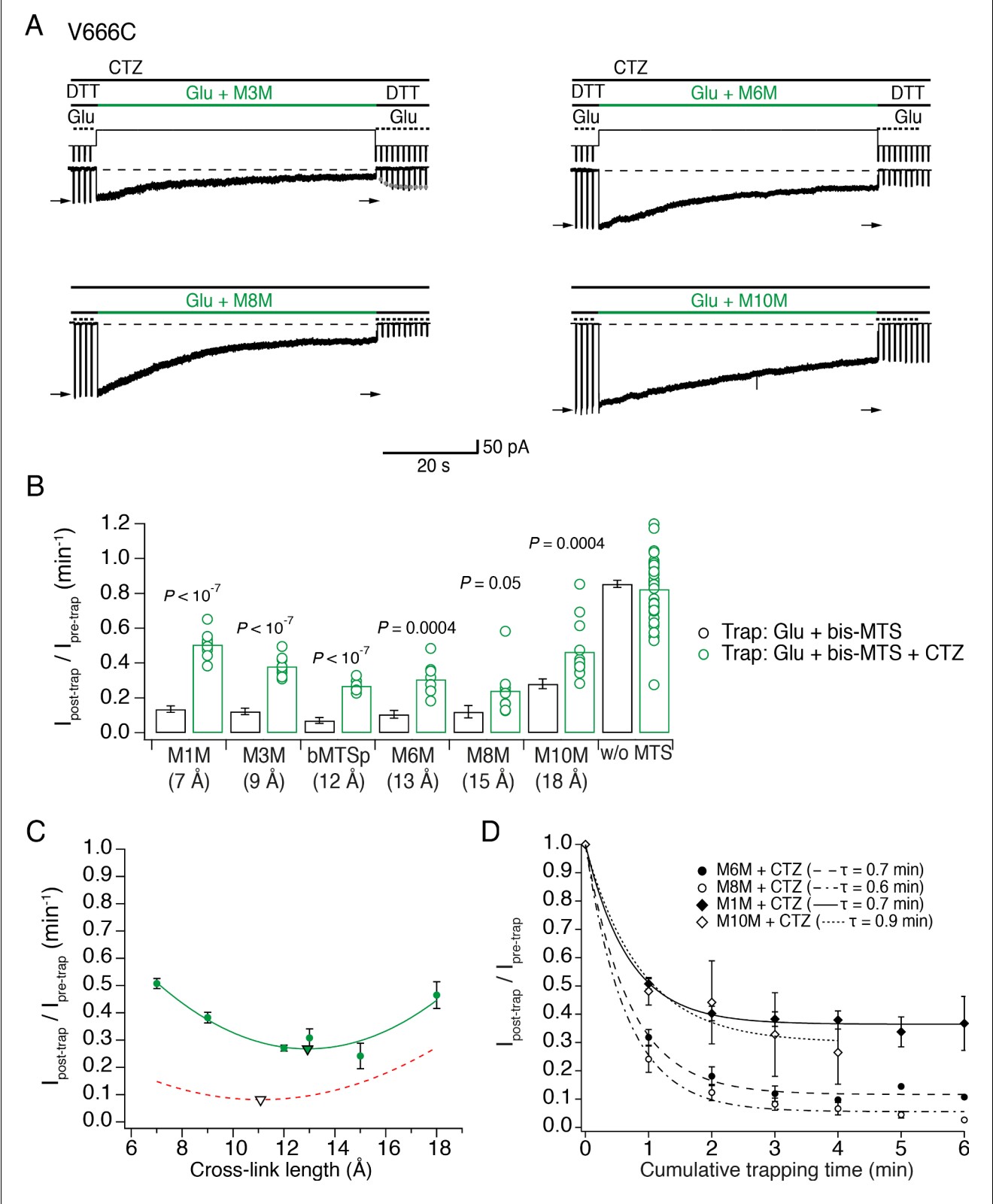

**Figure 5.** The active LBD layer is dilated. (**A**) Current traces of trapping active V666C receptors. Cyclothiazide (CTZ) was included at 100 µM throughout the experiment to block desensitization. The duration of the trapping pulse is indicated in green. For M3M, gray dots indicate recovery from trap immediately after the trap (τ = 2.0 ± 0.2 s, n = 8). Post- and pre-trap current amplitude was determined from the control pulses (arrows). (**B**) Summary of bis-MTS trapping (1 min) of desensitizing (black) and non-desensitizing V666C receptors (green). Desensitizing data are from **Figure 2E**. p values

*Figure 5 continued on next page*

*Figure 5 continued*

compare the desensitizing and non-desensitizing condition. For p values comparing effects with and without bis-MTS and between the cross-linkers, see *Figure 5—source data 1*. Bis-MTS length is indicated in brackets; 'w/o MTS': without MTS. (**C**) Trapping profile of active V666C receptors after the first minute of exposure fit with parabola (green line), reaching the minimum (green triangle) at (13, 0.3). Trapping profile of desensitized receptors is shown as a red, dashed line (parabola with minimum at 11 Å, black triangle). (**D**) Dependence of the trapping time on the length of the bis-MTS. V666C receptors were exposed up to six times to the trapping protocol described in (**A**). Current reduction was determined after each 1-min application to a bis-MTS. Current decay was described with a monexponential fit (τ in brackets).

DOI: https://doi.org/10.7554/eLife.40548.012

The following source data and figure supplement are available for figure 5:

**Source data 1.** Statistics of trapping active V666C receptors with different bis-MTS cross-linkers; data from *Figure 5B*.

DOI: https://doi.org/10.7554/eLife.40548.014

**Figure supplement 1.** No silent modification of active V666C receptors.

DOI: https://doi.org/10.7554/eLife.40548.013

with GSG1L auxiliary proteins (*Figure 1C*, PDB: 5VHZ [*Twomey et al., 2017b*]). If auxiliary subunits keep the LBD layer more compact, we reasoned that their presence should also limit the effects of longer bis-MTS cross-linkers. For example, the longest cross-linker (M10M) could be expected to have less effect on the desensitized V666C receptors in the presence of Stg.

To test this hypothesis, we repeated the trapping experiments on complexes of AMPA receptors with Stg (*Figure 6*). GluA2 and Stg were co-expressed and association of complexes was assessed by measuring the ratio of kainate current over glutamate current (KA/Glu). The relative efficacy of the partial agonist kainate is known to be higher for GluA2-Stg complexes than for GluA2 alone, making it a good marker of GluA2-Stg association (*Tomita et al., 2005*; *Shi et al., 2009*). After establishing formation of the GluA2 V666C-Stg complexes in the patch, we proceeded with the trapping protocol that exposed complexes to glutamate and a bis-MTS cross-linker (1 μM) for 1 min (*Figure 6A–B*), as described previously. No blocker of desensitization was added and the receptors were allowed to desensitize freely. Stargazin reduces desensitization of AMPA receptors so the crosslinking represents trapping across a mixture of active and desensitized states.

The trapping results summarized in *Figure 6C* show that the presence of auxiliary subunit Stg apparently protected V666C receptors from cross-linking by bis-MTS. Indeed, following trapping, a robust response was preserved, and could not be overcome by longer trapping intervals (*Figure 6—figure supplement 1A–B*). Based on published structures, the decrease in the trapping extent was expected for M10M (*Figure 6C* and *Figure 6—source data 1*), but not for shorter bis-MTS. This result indicates the presence of alternative conformations that have a more compact extracellular layer or that otherwise render cysteines inaccessible to modulation by bis-MTS.

To test whether bis-MTS cross-linkers perhaps act on non-complexed V666C receptors only, without affecting V666C-Stg complexes, we measured the KA/Glu current ratio before and after the bis-MTS trap for a series of patches. We reasoned that if only non-complexed V666C receptors were being modified, the glutamate-activated current should reduce, but the kainate current (which is almost entirely carried by GluA2-Stg complexes) should not. Therefore, preferential trapping of non-complexed V666C mutants should lead to an increase in KA/Glu ratio. As shown in *Figure 6E*, the KA/Glu ratio was not affected (before trap: 0.46 ± 0.03; after trap: 0.43 ± 0.02, n = 13; *p* = 0.2, paired randomization test), indicating V666C-Stg complexes were being modified by bis-MTS cross-linkers.

## Stargazin maintains active receptors in a compact arrangement

The trapping profile of V666C-Stg complexes (orange in *Figure 6D*) reflects the partial protection from trapping in the presence of Stg for all cross-linker lengths, but its overall shape is practically superposable onto the trapping profile of desensitized V666C receptors without Stg (red, dashed line in *Figure 6D*). Strikingly, the two curves reach their minimum at the same point of 11 Å (triangles in *Figure 6D*) and have identical curvature. This indistinguishable length dependence indicated that the trapping of V666C-Stg complexes came primarily from trapping desensitized receptors, and that the active complexes of V666C-Stg might be untouched by bis-MTS cross-linkers.

To test this hypothesis, we repeated the trapping protocol in the presence of CTZ, to block desensitization of V666C-Stg complexes. Two cross-linkers were tested in this condition: M1M, the

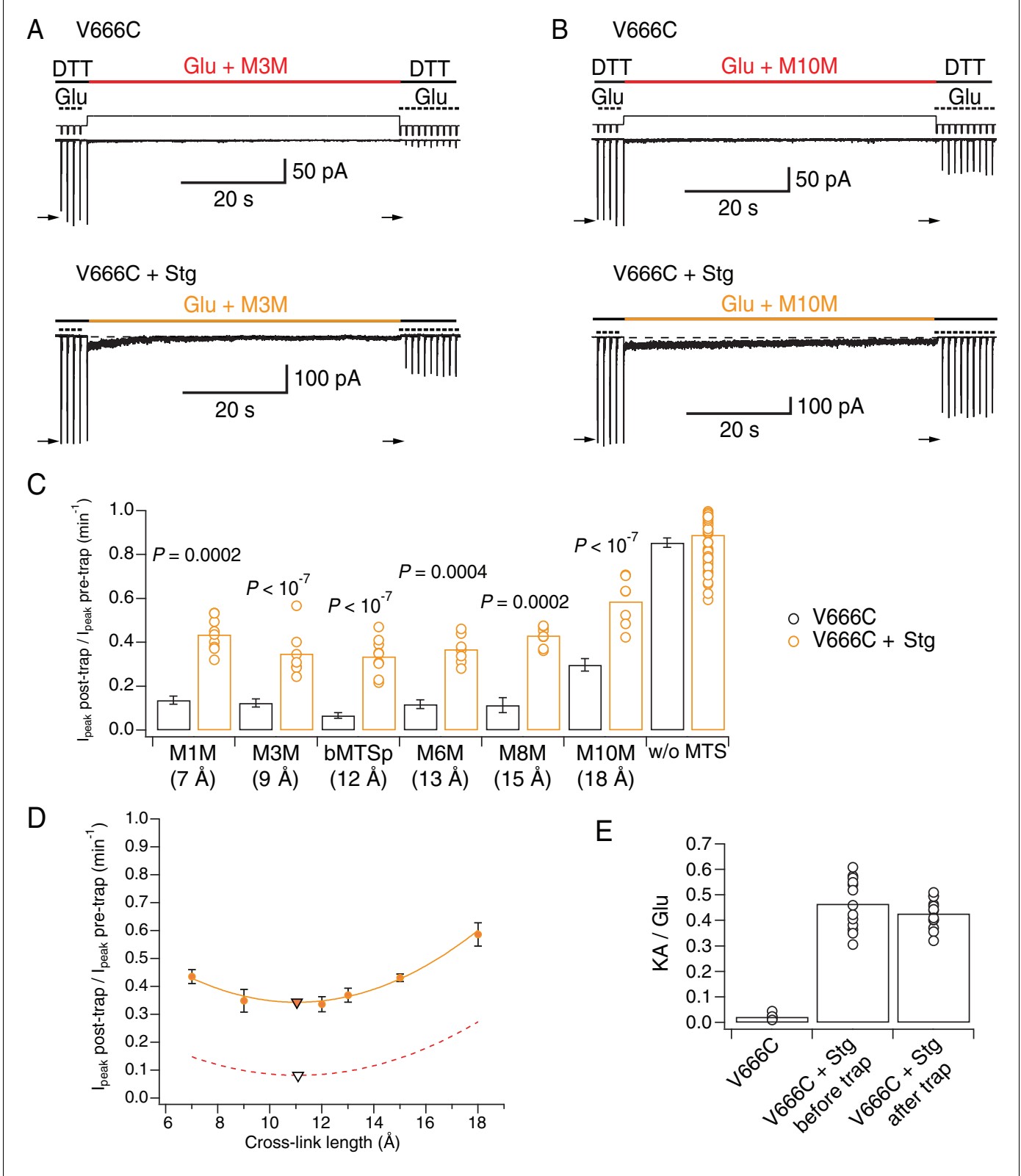

**Figure 6.** Stargazin attenuates effects of bis-MTS cross-linkers. (**A**) Current traces of V666C trapping in M3M, without (top) and with Stargazin (Stg; bottom). Legend is the same as in *Figure 2*, with a trapping pulse shown here in orange. (**B**) Same as in (**A**), but for M10M. (**C**) Trapping effects for V666C without (black) and with Stg (orange). Post- and pre-trap peak current was determined from the control pulses (arrows in (**A**) and (**B**)). Data without Stg are the same as in *Figure 2D*. p values compare the effects of the respective cross-linker with and without Stg. For statistics vs. w/o MTS

*Figure 6 continued on next page*

*Figure 6 continued*

and between cross-linkers, see *Figure 6—source data 1*. (D) Trapping profile of desensitizing V666C + Stg complexes. The data were fit with a parabola (orange line); the fit reaches minimum (orange triangle) at (11, 0.3). Trapping profile of desensitized receptors without Stg is shown as red, dashed line (parabola with minimum at (11, 0.1); black tringle). (E) The kainate/glutamate (KA/Glu) peak current ratio was determined for each patch before and after trapping with a bis-MTS (similar to the experimental design in *Figure 3A–C* with 1 mM KA and 10 mM Glu in 1 mM DTT). V666C + Stg KA/Glu measurements before and after trap shown here are paired recordings, pooled for various bis-MTS cross-linkers.
DOI: https://doi.org/10.7554/eLife.40548.015

The following source data and figure supplement are available for figure 6:

**Source data 1.** Statistics of trapping V666C receptors with Stargazin (Stg) with different bis-MTS cross-linkers; data from *Figure 6C*.
DOI: https://doi.org/10.7554/eLife.40548.017

**Figure supplement 1.** Cross-linking times depend on the length of the bis-MTS cross-linker, functional state of the receptor and the presence Stargazin.
DOI: https://doi.org/10.7554/eLife.40548.016

shortest one, and M8M, which had one of the strongest trapping effects on active V666C receptors without Stg (*Figure 5B–C*). As predicted, neither bis-MTS cross-linker had any effect on activated GluA2-Stg complexes (M1M: $p = 0.1$, $n = 9$; M8M: $p = 0.2$, $n = 9$; paired recordings with and without bis-MTS) (*Figure 7*). Over very long exposures, M8M could induce irreversible inhibition, presumably from residual desensitization (*Figure 6—figure supplement 1*). This result confirmed that partial trapping observed for desensitizing V666C-Stg mainly derived from bis-MTS cross-linkers accessing desensitized complexes.

We next wondered if the protection from trapping observed for active GluA2-Stg complexes could, at least partly, be explained by reduced accessibility of cysteines during exposure to bis-MTS, perhaps because the Cys666 residues are buried against other subunits or oriented such that the sulfhydryl groups are inaccessible for cross-linking. We mined possible conformations of the active LBD tetramer (PDB: 5WEO) using a coarse docking approach, allowing each active dimer to move in the membrane plane, and allowing rotation of one dimer with respect to the other around axes parallel and perpendicular to the active dimer interface (*Figure 7D and E*). Interestingly, docked structures with low Cys666 accessibility segregated into two classes, a loose arrangement where Cys666 were physically close but sterically hindering each other (with the centre of mass separation of the two active dimers comparable to 5WEO, for example models m17 and m32 in *Figure 7D*), and a tighter arrangement of the LBDs with V666C close to helix K (with the centre of mass separation of the two active dimers smaller than in 5WEO, for example m223 and m265 in *Figure 7D*), not unlike the structure of the V666C mutant LBD bound with fluorowillardine (5JEI) (*Salazar et al., 2017*). In that structure, additional shielding of the V666C side chain is afforded by its being shielded by neighboring structural elements.

Thus, docked structures lead us to hypothesize that active GluA2-Stg complexes could avoid cross-linking not just by adopting more compact LBD arrangements, but also by preferring conformations which shield V666C SG groups.

## Discussion

Technical advances in cryo-electron microscopy have revolutionized the study of membrane proteins, and the supply of structural information is greater than ever before. However, as the catalogue of images swells, the need to relate their geometry to dynamics becomes more pressing. In case of AMPA receptors, full-length structures in complexes with various ligands and auxiliary subunits have been published. Here, we have used a classical crosslinking approach allied to rapid perfusion to measure distances up to 18 Å in AMPA receptor extracellular domains across different functional states and in the time domain.

Bis-MTS cross-linkers have been used previously to aid identification of movements underlying state transitions in AMPA and NMDA receptors (*Armstrong et al., 2006*; *Tajima et al., 2016*). Here, we report effects with a strong dependence on the target site, length of the cross-linker, functional state of the receptor and the presence or absence of auxiliary subunits. Since most bis-MTS cross-linkers are alkyl chains and hence flexible, they can cross-link pairs of Cys residues closer to each other than the extended length of the cross-linker. For example, M10M could cause the same effect

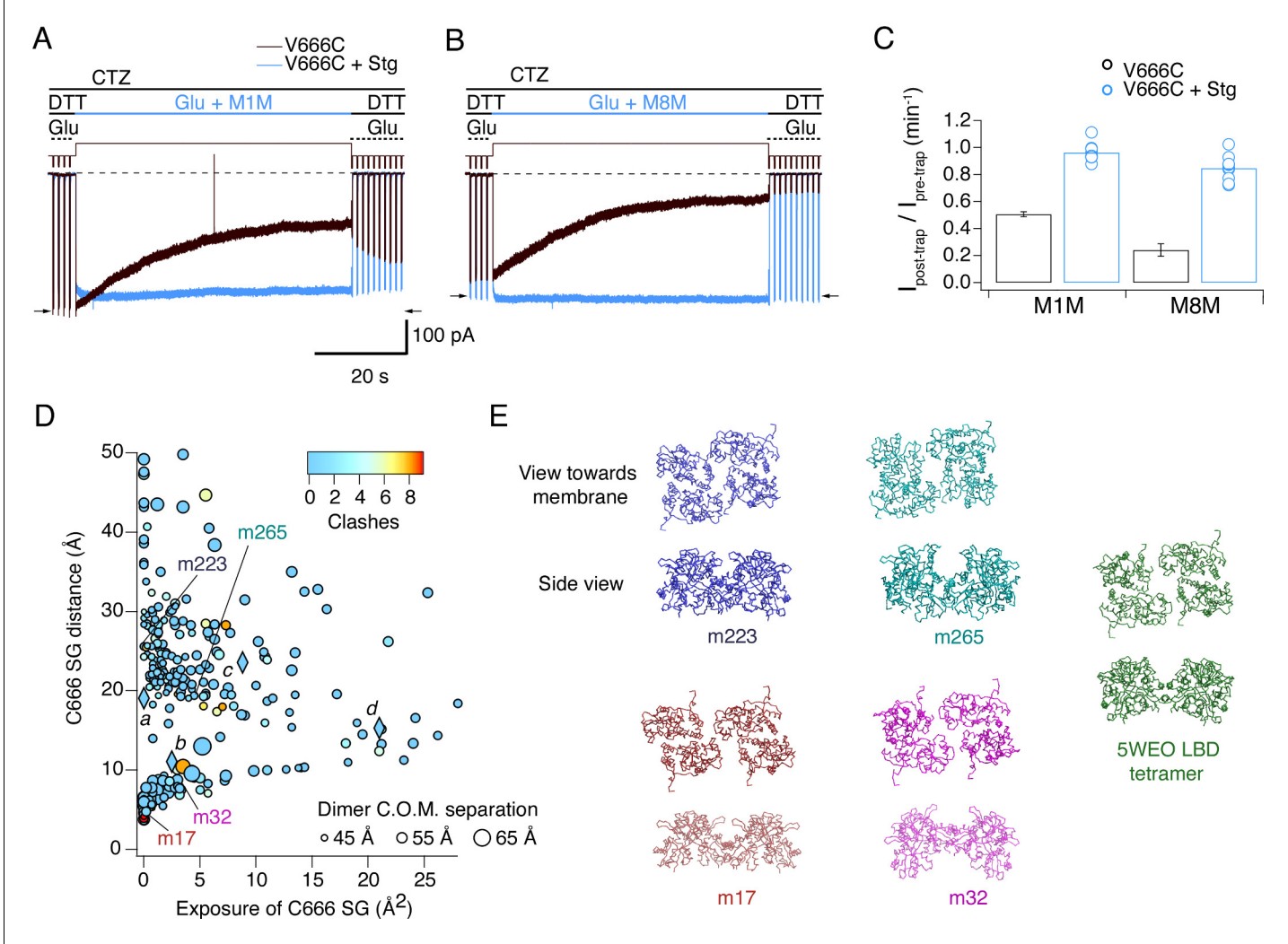

**Figure 7.** Stargazin blocks access to bis-MTS cross-linkers in the active state. (**A**) Traces showing trapping by M1M in the active state in the absence (black) and presence (blue) of Stargazin (Stg). Legend is the same as in *Figure 2*, with a trapping pulse shown here in blue. Desensitization was blocked by CTZ (100 µM) present throughout the experiment. Glutamate (Glu) was 10 mM and DTT 1 mM. (**B**) Same as in (**A**), but for M8M cross-linker. (**C**) Summary of the trapping effects for M1M and M8M in the active state in the absence (black) and presence (blue) of Stg. Pre- and post-trap current was determined from control pulses as indicated by arrows. (**D**) Putative compact structures that could protect from bis-MTS modification. Results of 268 runs of rigid body docking of LBD dimers against each other, to minimize Cys666 access in subunit A and mimic the protection from crosslinking. Results from known structures (5JEI, 4YU0 loose, 4YU0 tight and 5WEO, diamonds *a-d*) and four of the generated models – m17, m32, m223, and m265 - segregate into two classes, similar to the loose (SG-SG distance between subunits < 10 Å) and tight arrangements (SG-SG distance > 20 Å, C666 SG buried close to helix K) (*Baranovic et al., 2016*). The symbol size corresponds to the dimer centre of mass separation and the color to the number of atom clashes (distance < 2.2 Å.) For reference, dimer centre of mass separations of known structures are: 5JEI, 44 Å; 4YU0 loose, 47 Å; 4YU0 tight, 47 Å and 5WEO, 52 Å. Note that m17 is selected from a cluster of models with zero clashes (cyan), but adjacent in the graph to two other models with clashes (red circles). (**E**) LBD arrangements in four models marked on the graph in panel D and the original seed for each optimization, the glutamate bound LBD tetramer from 5WEO full-length GluA2 structure with Stg.

DOI: https://doi.org/10.7554/eLife.40548.018

when bound to two Cys that are 7 or 15 Å apart, acting as a universal inhibitor for a range of Cys distances. For this reason, we used bis-MTS ranging from 7 to 18 Å in length (M1M – M10M). Our results show an optimum (inhibitory) effect of the cross-linkers at position V666C, with M10M not producing as strong effect as shorter cross-linkers. This is inconsistent with the interpretation of the universal inhibitor and suggests bis-MTS cross-linkers exert their effect mainly by preventing the engineered residues from separating more than the extended length of the cross-linker. We also tested a rigid cross-linker, bMTSp, whose length matches closely that of flexible M6M (12 and 13 Å,

respectively). This places a lower limit on separation of V666C Cys at ~12 Å. The only difference we observed between bMTSp and M6M was with intra-dimer K493C mutant, with M6M blocking desensitization more effectively than bMTSp, perhaps indicating an angle between the K493C side chains that was more easily accommodated by the flexible chain.

All bis-MTS cross-linkers, when applied to the mutants at the inter-dimer LBD interface, inhibited the current, albeit to a different degree depending on the functional state of the receptors. Whereas cross-linking of the desensitized states should result in current inhibition, cross-linking of an open channel would be expected to result in more active receptors. However, to date all inter-dimer traps have been inhibitory. Inhibition seems independent of the nature of the constraint, whether disulphide bonds (*Plested and Mayer, 2009*; *Yelshanskaya et al., 2016*), metal bridges (*Lau et al., 2013*; *Baranovic et al., 2016*), flexible or rigid bis-MTS cross-linkers (this study). In addition, a structure of the full-length GluA2 receptor carrying a gain-of-function mutation and in complex with a partial agonist, blocker of desensitization and a peptide constraining the LBD dimers resulted in a closed channel (*Chen et al., 2014*). Cysteine crosslinks within dimers inhibit the gating of kainate receptors, even when blocking desensitization (*Daniels et al., 2013*). Whereas the peptide binding, zinc and disulphide bridges only form under strict geometrical requirements, possibly incompatible with a stably open pore, the same cannot be said for bis-MTS cross-linkers. The cross-linkers are flexible and therefore do not restrict the trapped LBD tetramer to a single geometry. The inhibitory effect of the crosslinkers is not universal – at the active intra-dimer interfaces we could potentiate receptors as shown previously (*Armstrong et al., 2006*). Structural models of full-length AMPA receptors in various states of activity do not explain why every restraint of the LBD dimers leads to a decrease in activity. Either the correct sites for potentiation have not yet been found, or this observation reflects a mechanistic detail of receptor activation. The mechanism behind this inhibition remains frustratingly unclear, limiting the interpretation of the cross-linking approach used in this study.

In *Figure 8A–B*, the trapping profiles of V666C receptors with the available structural models of GluA2 in the equivalent condition are compared. The shallow trapping profile of desensitized AMPA receptors indicates a structural ensemble of conformations broadly in agreement with multiple desensitized states (*Meyerson et al., 2014*; *Robert and Howe, 2003*). Even though the LBD dimer-dimer separation observed in the model (> 18 Å, EMDB: 2688) was not tested with the bis-MTS cross-linker of the corresponding length, extrapolation of the trapping profile indicates such conformations are available to desensitized AMPA receptors. However, the longest cross-linker (18 Å) was the slowest one to act on desensitized receptors, indicating that this conformation, and more 'open' ones, are not readily available for cross-linking, but slowly get populated during prolonged exposures to agonist. The longest cross-linker, M10M, was also the slowest one to act in the active state. There, the strongest cross-linking effect was achieved by the 12–15 Å-long bis-MTS (green in *Figure 8B*). This suggests inter-dimer V666C residues are closer together in the active state than 19 Å, predicted by the structural model (PDB: 4UQ6). The extracellular layer was also seen in an 'opened' arrangement in NMDA receptors (*Zhu et al., 2016*), with the caveat that these structures were antagonist-bound and therefore not equivalent to our observations.

The presence of Stg lead to universal attenuation of the cross-linking effect, for all bis-MTS lengths and both functional states, desensitized and active (orange and blue in *Figure 8A and B*, respectively). The longest crosslinker (M10M) produced the least trapping, even following long exposures (~3 min). Thus, the LBD layer could not be modulated by bis-MTS cross-linkers for majority of these receptors. This result chimes with structural and FRET experiments (*Shaikh et al., 2016*), where the presence of Stg made the LBD and ATD layer more compact. In the desensitized state, the shape of the trapping profile is the same, with or without Stg (*Figure 8A*), with both parabola reaching a minimum point at 11 Å (triangles in *Figure 8A*). This is in excellent agreement with the structure of desensitized GluA2-Stg complex (PDB: 5VOV). In the active state with Stg, our results deviate from the available full-length active structures. Both short and long (7 or 15 Å) bis-MTS reagents failed to inhibit currents in the first minute of trapping, indicating more compact LBD arrangements over this timescale than obtained in the structures. In other words, if structures can be obtained in a time-resolved manner in milliseconds following glutamate exposure (*Unwin, 1995*), our results suggest they should be more compact than structures solved to date.

The protection from trapping in our cross-linking experiments most likely has multiple origins. Although 'protection' could result from a persistent long-distance separation, outside the range of

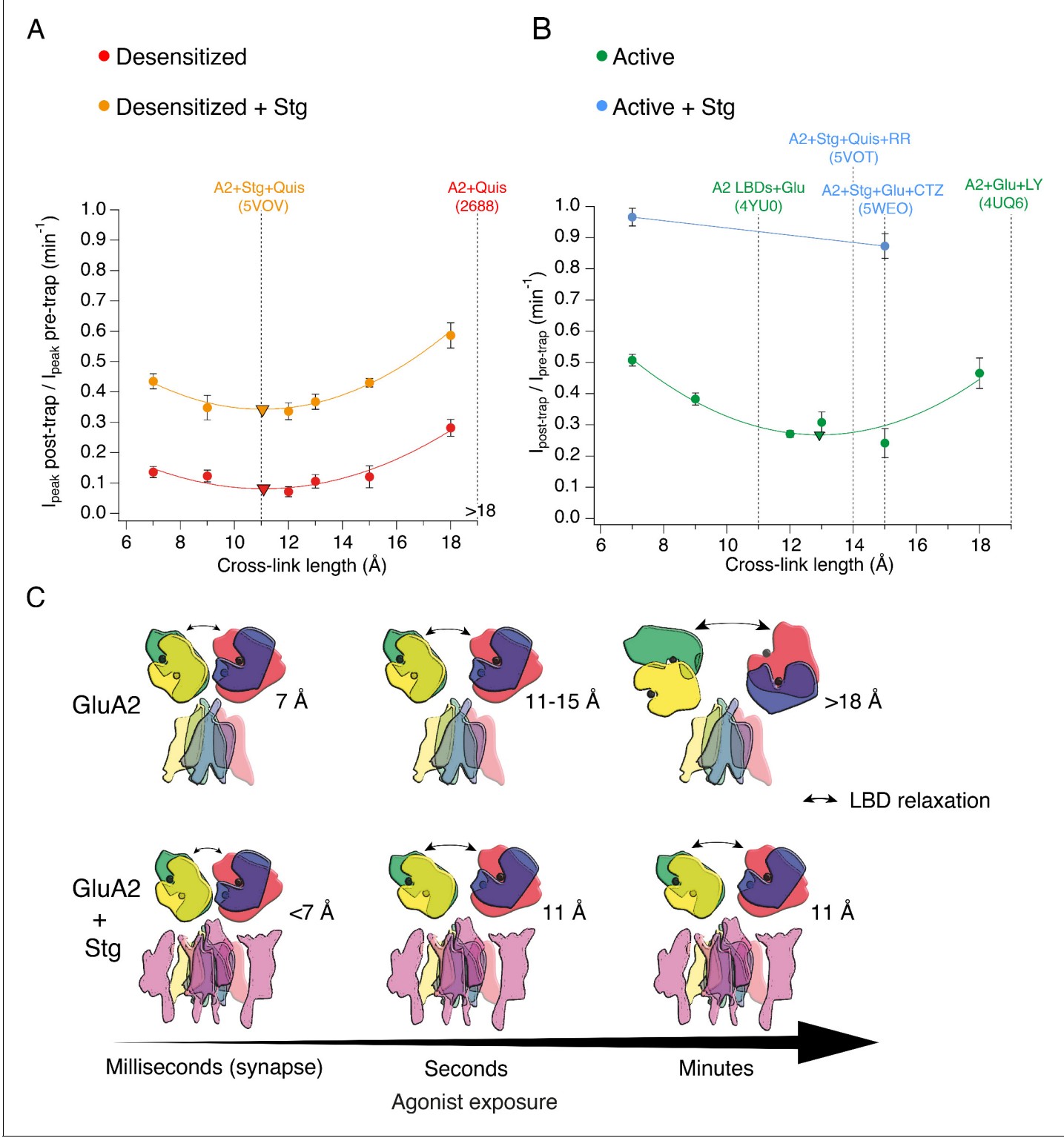

**Figure 8.** The presence of Stargazin and short agonist exposures keep the agonist-bound LBD layer compact. (**A**) Trapping profiles for V666C in desensitized states, without (red) and with (orange) Stargazin (Stg). Solid lines are fit parabola, with local minima indicated as triangles. The right handside of the x-axis is marked as > 18 Å, as the resolution of the only available desensitized structure without auxiliary subunits is too low to measure residue distances. Vertical, dashed, black lines indicate V666C sulfhydryl separation in the respective structure (PDB accession codes in brackets). Names of structures are color-coded same as the trapping profiles. (**B**) Same as in (**A**), but for active state without (green) and with Stg (orange). 4YU0 is a structure of soluble, isolated LBDs. Quis: quisqualate (full agonist), RR: (R, R) −2b (*Kaae et al., 2007*) and LY: LY451646 (both desensitization

*Figure 8 continued on next page*

Figure 8 continued

blockers). (C) Schematic summarizing the bis-MTS cross-linking results. Color-code same as in *Figure 1*; ATDs are omitted. Black spheres represent bound glutamate. Upper row: in the absence of Stg, GluA2 receptors access 'open' LBD conformation (black arrows) upon long exposures to agonist. Bottom row: presence of Stg limits 'opening' of the LBD layer even at longer agonist exposures.

DOI: https://doi.org/10.7554/eLife.40548.019

The following figure supplement is available for figure 8:

**Figure supplement 1.** Cartoon depicting cross-linking (paperclips) of A-C subunits (green and blue), while B-D subunits are left free (red and yellow).

DOI: https://doi.org/10.7554/eLife.40548.020

the crosslinkers, several observations and common sense speak against this possibility. First, we previously showed that active receptors could be trapped by zinc bridges in compact arrangements (*Baranovic et al., 2016*). Second, the long distance would have to be maintained throughout the exposure, because any transit between compact and dilated arrangements must pass through intermediate separations, allowing crosslinkers to span the gap, assuming extremely rapid reactivity of the bis-MTS compounds (*Kenyon and Bruice, 1977*). Our reaction rates are close to the maximum expected ($10^5$ $M^{-1}s^{-1}$; *Liu et al., 1997*). Third, even the most dilated structures are in the range of crosslinker lengths that we used. Fourth, the mixed trapping condition for GluA2-Stg complexes in the absence of CTZ apparently supports desensitized state trapping over a wide range of geometries but no additional active state trapping. We cannot discount specific, state-dependent protection from crosslinking, produced by a unique, closed-cleft LBD conformation provoked by Stg in the active state, but it seems to us unlikely. We reasoned that protection from crosslinking could, at least partly, be accounted for by reduced accessibility of cysteine residues during exposure to bis-MTS. Docking data in *Figure 7D–E* show that conformations which reduce the accessibility of the V666C side chains are indeed accessible to the active LBDs. These arrangements are more compact than the available structures of the full-length receptor in the active state (*Chen et al., 2017*; *Twomey et al., 2017a*).

No matter how avid and complete trapping is, a possible source of receptor activity after trapping is the activity of free (non-crosslinked) subunits (*Figure 8—figure supplement 1*). Based on the GluA2 structural models, bis-MTS compounds cross-link subunits proximal to the central twofold symmetry axis at the LBD level (subunits A and C), while sparing the distal subunits, B and D (*Figure 1A–C*). In AMPA receptors without Stg, with desensitization blocked, the activity of only two subunits produces subconductance openings with short open times (*Rosenmund et al., 1998*). But the presence of Stg (*Coombs et al., 2017*; *Zhang et al., 2014*) increases the current carried by receptors with one or two active subunits. In our experiments, Stg had an indefatigable effect of maintaining receptor activity in the limit of long exposures to bis-MTS. The effect of Stg was inordinately large, with a maximum effect of the crosslinking leaving approximately half the receptor activity unscathed. Can the two non-crosslinked subunits (*B* and *D*) produce this level of activity? A simple back-of-the-envelope calculation suggests this effect is larger than expected. Single channel recordings of GluA2 with Stg reveal that the conductance from two subunits, with desensitization blocked, should be on average about 40% of a full opening but with $P_{open} < 1$ (*Coombs et al., 2017*). Therefore, residual activity of the non-crosslinked subunits is higher than expected, unless the B and D subunits have a predominant role in driving channel gating (as proposed from structural studies [*Sobolevsky et al., 2009*]).

Physiological activation and desensitization of AMPA receptors takes place on a millisecond timescale. The fast gating contrasts desensitized states trapped by bis-MTS cross-linkers that take tens of minutes to recover. It seems reasonable to assume that the degree of stabilization by different crosslinkers is similar, and that the difference in stability comes from the states themselves. This idea is supported by the cut-off that we observe – for crosslinkers longer than 10 Å, crosslinking in desensitized states is irreversible over 10 min (also in fivefold higher concentration of the reducing agent). It is possible that upon cross-linking, V666C residues adopt conformations that make them inaccessible to the reducing agent, but this would then have to be highly specific for bis-MTS reagents longer than 9 Å and state-specific (due to much faster recovery rates of active than desensitized receptors after trapping in M1M or M3M). In addition, in reducing conditions, AMPA receptors recover from disulphide crosslinking at the same sites in hundreds of milliseconds (*Lau et al., 2013*; *Salazar et al., 2017*). Assuming that the length of the bis-MTS cross-linkers reflects level of structural

rearrangements and following the trend plotted in *Figure 4H*, this suggests that AMPA receptors at synapses, exposed to brief glutamate transients and in complex with auxiliary subunits, are unlikely to undergo extreme conformational changes. Considering that all of the experiments presented here were done with over-expressed receptors and auxiliary subunit Stg, it is possible that V666C-Stg complexes had variable stoichiometry of association, depending on the expression level. This could impact how Stg affects conformational dynamics of the receptors, but a detailed investigation of this is beyond the scope of this work. We note, however, that the presence of two copies of auxiliary subunit GSG1L were sufficient to keep the LBD layer compact in structural experiments (*Twomey et al., 2017b*).

Long agonist exposures of minutes to hours are standard in structural biology experiments and could, thus, contribute to the prevalence of more 'open' conformations in full-length structures without auxiliary subunits (*Figure 8C*). However, long exposures to agonists are at odds with synaptic conditions where AMPA receptors see glutamate on a millisecond timescale, before it is actively cleared by transporters (*Clements, 1996*). Furthermore, any large structural rearrangements of the extracellular domains would need to be accommodated by the crowded synaptic environment (*High et al., 2015*; *Tao et al., 2018*) and potential presynaptic interaction partners (*Saglietti et al., 2007*; *Elegheert et al., 2016*). When in complex with auxiliary subunits, no functional state of AMPA receptors necessitates large domain movements, and compact arrangements of the extracellular layer can sustain the gating process. We conclude that extracellular domains of synaptic AMPA receptors are unlikely to undergo large structural rearrangements during synaptic transmission and instead work in a fairly compact conformational regime, unless faced with long exposures to glutamate in pathological conditions.

# Materials and methods

## Key resources table

| Reagent type (species) or resource | Designation | Source or reference | Identifiers | Additional information |
|---|---|---|---|---|
| Cell line (*Homo sapiens*) | HEK293 | RRID: CVCL_0045 | ACC No. 305 | Obtained from Leibniz Forschungsinstitut DSMZ (Deutsche Sammlung von Mikroorganismen und Zellkulturen GmbH, Germany). |
| Transfected construct (*Rattus norvegicus*) | GluA2; GluA2flip | Other | ID_GenBank: M38061.1 | Gift from Mark Mayer (NIH); vector: pRK5 with IRES eGFP, unedited (Q) at the pore site, signal peptide: 21 amino acid. |
| Transfected construct (*Mus musculus*) | Stargazin; Stg | Other | ID_GenBank: NM_007583.2 | Gift from Susumu Tomita; vector: pRK5 with IRES dsRed (dsRed_Max Addgene plasmid: 21718). |
| Commercial assay or kit | Nucleobond Xtra Midi EF | Macherey and Nagel | 740420.5 | Plasmid DNA preparation kit. |
| Chemical compound, drug | Bifunctional methanethiosulfonate cross-linkers; bis-MTS | Toronto Research Chemicals, North York, Canada | M1M: M258800; M3M: P760350; M6M: H294250; bMTSp: P193250; M8M: O235850; M10M: D210875 | Obtained as powder, stock prepared in DMSO, used at final concentration of 1 μM or 50 μM, as indicated. |

*Continued on next page*

*Continued*

| Reagent type (species) or resource | Designation | Source or reference | Identifiers | Additional information |
|---|---|---|---|---|
| Chemical compound, drug | Monofunctional methane thiosulfonate linker; MTSEA | Toronto Research Chemicals, North York, Canada | MTSEA: A609150 | Obtained as powder, stock prepared in DMSO, used at final concentration of 1 µM. |
| Chemical compound, drug | Cyclothiazide; CTZ | Hello Bio, Bristol, UK | HB0221 | Obtained as powder, stock prepared in DMSO, used at final concentration of 100 µM. |
| Chemical compound, drug | Kainate; KA | Abcam plc, Cambridge, UK | ab120100 | Obtained as powder, stock prepared in Ringer, used at final concentration of 1 mM. |
| Software, algorithm | Docking | Source: https://github.com/aplested/cystance (copy archived at https://github.com/elifesciences-publications/cystance); this paper (*Plested, 2018*) | | Python script for PyMOL and CCP4. |
| Software, algorithm | Randomization test | Source: https://github.com/aplested/DC-Stats (copy archived at https://github.com/elifesciences-publications/DC-Stats); DC-Stats suite (*Plested and Lape, 2018*) | | 105 iterations performed for each test. |

## Molecular biology

In all experiments, the unedited (Q586) GluA2flip version of rat GluA2 gene was expressed from the pRK5 vector. Amino acid numbering refers to the mature receptor assuming a signal peptide of 21 amino acids in length. As a marker of transfection, eGFP was expressed from the same vector, downstream from an internal ribosomal entry sequence (IRES). Mouse Stargazin gene (a kind gift from Susumu Tomita) was expressed from a separate pRK8 vector containing IRES-dsRed (*Carbone and Plested, 2016*). All mutations were introduced by overlap PCR and confirmed by double-stranded sequencing.

## Cell culture and transfection

GluA2 constructs were expressed transiently in HEK293 cells using calcium-phosphate precipitation or PEI method as described previously (*Baranovic et al., 2016*; *Riva et al., 2017*). HEK293 cells were obtained from the Leibniz Forschungsinstitut DSMZ (Deutsche Sammlung von Mikroorganismen und Zellkulturen GmbH, Germany) ACC no. 305 (RRID: CVCL_0045) and tested negative for mycoplasma. Cells were maintained in MEM Eagle medium (PAN-Biotech GmbH, Aidenbach, Germany) supplemented with 10% (v/v) fetal bovine serum and antibiotics (penicillin (100 U/mL) and streptomycin (0.1 mg/mL; PAN-Biotech).

For transfections, 2–3 µg of DNA was transfected per 35 mm dish and cells were washed after 6–8 hr. Recordings were performed 24–72 hr after the transfection at room temperature. For transfections with Stargazin, Stargazin DNA was co-transfected with GluA2 DNA at 2:1 mass ratio and after the transfection, cell medium was supplemented with 40 µM NBQX to reduce Stargazin-induced cytotoxicity.

## Solutions

Chemicals were obtained from Sigma Aldrich (Munich, Germany), Abcam plc (Cambridge, UK) and Hello Bio (Bristol, UK). MTS compounds were obtained from Toronto Research Chemicals (North York, Canada).

The internal (pipette) solution for recordings without Stargazin contained (mM): 115 NaCl, 1 $MgCl_2$, 0.5 $CaCl_2$, 10 NaF, 5 $Na_4$BAPTA, 10 $Na_2$ATP, 5 HEPES, titrated to pH 7.3 with NaOH. For recordings with Stargazin, the internal solution was slightly modified: 120 NaCl, 0.5 $CaCl_2$, 10 NaF, 5 $Na_4$BAPTA, 5 HEPES and 0.05 spermine, pH 7.3. The external recording solution in all experiments

contained (mM): 150 NaCl, 0.1 MgCl$_2$, 0.1 CaCl$_2$ and 5 HEPES, pH 7.3. Different drugs were added to the external solution as needed. Glutamate was always applied at 10 mM and DL-dithiothreitol (DTT) at 1 mM. Cyclothiazide (CTZ) and kainate (KA) were thawed from stock solutions on the day of the experiment. Final CTZ and KA concentrations in all experiments were 100 µM and 1 mM, respectively.

All MTS compounds were obtained as powder. Although monofunctional MTS reagents are known to be highly reactive and unstable in aqueous solutions (*Kenyon and Bruice, 1977*), this information is lacking for bifunctional cross-linkers used in this study. Hence, we took special care to minimize exposure of bis-MTS compounds to oxidizing (aqueous) solutions (*Takatsuka and Nikaido, 2010*). The powder was dissolved in DMSO, aliquoted and kept on ice on the day of the experiment. Once a stable patch recording was obtained, an aliquot was dissolved in external solution to a final concentration of 1 µM and applied to the patch. This way, aqueous bis-MTS solutions were on average 2–3 min old at the moment of application. Each MTS stock was tested with GluA2 wild-type receptors and K493C receptors as negative and positive controls, respectively. The final MTS concentration of 1 µM was chosen based on previous work (*Sobolevsky et al., 2003*; *Yelshansky et al., 2004*). We avoided higher concentrations of MTS compounds due to potential cross-reactivity and chaining effects.

## Patch clamp electrophysiology

Ligands and drugs were applied to outside-out patches via a custom made 4-barrel glass (VitroCom, USA) mounted to a linear piezo–electric wafer (PiezoMove P-601.4, PI, Germany) (*Lau et al., 2013*). Two barrels were perfused with control solutions and the third barrel with the trapping solution, as described below. All patches were voltage clamped at −40 mV unless stated otherwise. Currents were low-pass filtered at 10 kHz (−3 dB cut-off, eight-pole Bessel filter) using an Axopatch200B amplifier (Molecular Devices, USA) and acquired with AxographX software (Axograph Scientific, Australia, RRID:SCR_014284) at 20 kHz sampling rate via Instrutech ITC-18 digitizer (HEKA, Germany). Current traces were digitally filtered at 1 kHz (low-pass) for presentation in figures.

To assess the effect of different bifunctional cross-linkers on AMPA receptors, the receptors were exposed to the cross-linker (1 µM) for 1 min. Before and after this trapping exposure, the current in the patch was tested with control pulses that contained only 10 mM glutamate, without the cross-linker and in the presence of DTT (1 mM) as a reducing agent (*Figure 2A–C*). Four control pulses before application of the cross-linker provided a measure of the patch current before any exposure to the cross-linker. Accordingly, (up to thirty) control pulses recorded after the MTS application, were used to assess any changes in the patch current imparted by the cross-linker treatment.

For recordings of GluA2 receptors co-expressed with auxiliary subunit Stargazin, care must be taken that GluA2 receptors are indeed associating with Stargazin. One strategy to minimize the presence of lone V666C receptors relies on the relief of spermine (polyamine) block at positive voltages imparted by complexation with Stargazin (*Carbone and Plested, 2016*). Although we have included spermine in the pipette solution and measured relieve of block for each patch, we did not perform recordings at positive voltages, as the currents were not stable enough during minutes-long trapping protocols. Instead, a change in kainate efficacy was used as a marker of GluA2-Stargazin association as described in the text.

## Analysis

Trapping effects were quantified as the ratio of the average current after the trap (determined from the second post-trap control pulse) and average current before the trap (determined from the four pre-trap control pulses; arrows in *Figure 2A–C*):

$$Active\,fraction = \frac{I_{post\,trap}}{I_{pre\,trap}}$$

In case of desensitizing receptors, peak current was measured and in the case of non-desensitizing receptors, steady-state current.

The trapping time of each cross-linker (at 1 µM) was determined from cumulative exposures to a bis-MTS of up to 6 min (six repetitions of the 1-min trapping protocol). After each application, current reduction was determined with respect to the initial current in the patch, before any trap. The

resulting current decay was described by a monoexponential fit in Igor Pro (v7.06, Wavemetrics, Lake Oswego, Oregon, USA, RRID:SCR_000325). This approach most likely overestimates the time M3M and M6M need to trap the V666C receptors, as majority of the current is inhibited within the first minute of exposure to bis-MTS (*Figure 2F*). For the experiments with 50 µM bis-MTS (M3M and M10M, *Figure 3*), where faster trapping rates are expected, the experiment was redesigned to measure the peak current inhibition every 50 ms, rather than in 1-min intervals.

To determine the rate of recovery from trapping by MTS cross-linkers, the number of post-trap control pulses was increased until full recovery was attained. An envelope of post-trap peak current responses was then created in Igor Pro and fit with a monoexponential. This approach was possible only for faster recovery rates, on the time scale of seconds, such as recovery of desensitized V666C receptors from trapping with M1M (*Figures 2B* and *4D*) and from direct V666C sulfhydryl cross-links (*Figure 4H*) and recovery of active V666C receptors from trapping with M1M and M3M (*Figure 5A*). With longer bis-MTS cross-linkers, the recovery time increased from seconds to minutes, making direct measurements of the recovery time from post-trap control pulses impractical. Instead, the experimental design was re-adjusted to allow measurements of long recovery times as described in *Figure 4A–C*. In brief, peak current in the patch was initially recorded with 100 ms jumps into glutamate in control conditions until it stabilized. Then, a trapping protocol was performed as described above, with control pulses before and after the trap. After the trapping protocol, the current in the patch was again monitored, for about 10 min, with fast control jumps into glutamate in order to follow any potential recovery of the peak current. In this time period, desensitized V666C receptors managed to recover only from trapping by M1M and M3M (*Figure 4D–G*), in which case the recovery was fit with a monoexponential.

Trapping profiles (*Figures 2G* and *6D*) were fit with a parabola in Igor Pro:

$$f(x) = K_0 + K_1(x - K_2)^2$$

where $K_1$ defines the curvature, $K_0$ the minimum effect and $K_2$ the x value at the minimum. Data points were weighted by the standard error of the mean for the fit.

## Computational docking

To investigate structures of the LBD tetramer that could preclude trapping by blocking access to the Cys666 SG moiety, we treated each dimer as a rigid body and subjected them to rotations and translations in the membrane plane. Python scripts (available at https://github.com/aplested/cystance; copy archived at https://github.com/elifesciences-publications/cystance) were written as a glue for molecular manipulations in PyMOL and CCP4 (RRID:SCR_007255) functions to measure geometry and exposure of the cysteine (AREAIMOL, NCONT (*Winn et al., 2011*)). For each run, trial arrangements that reduced Cys666 SG accessibility in subunit *A* whilst also keeping the dimers in close proximity (with minimal atom clashes) and maintaining physiologically plausible in-plane linker arrangements were retained as seeds for subsequent rounds, and the step size was reduced. Trial arrangements with more than 10 atom clashes (< 2.2 Å) were rejected. No refinement was done to eliminate spurious clashes from flexible surface residues. The optimization was ended when no further improvement was possible. Each search lasted about 5–10 min on a 2017 Macbook Pro.

All p values were determined by non–parametric randomization test (non–paired, unless stated otherwise), using at least $10^5$ iterations (DC-Stats suite: https://github.com/aplested/DC-Stats). Bars in graphs indicate mean and error bars SEM.

Structural models and related measurements were visualized and measured in PyMOL (v2.0, RRID:SCR_000305) (The PyMOL Molecular Graphics System, Version 2.0 Schrödinger, LLC). The length of bis-MTS cross-linkers was measured in ChemDraw Professional (PerkinElmer, USA.).

## Additional information

### Funding

| Funder | Grant reference number | Author |
| --- | --- | --- |
| H2020 European Research Council | Gluactive (647895) | Andrew J R Plested |

| Deutsche Forschungsge-meinschaft | NeuroCure EXC-257 | Andrew J R Plested |
|---|---|---|
| Deutsche Forschungsge-meinschaft | Heisenberg Professorship | Andrew J R Plested |

The funders had no role in study design, data collection and interpretation, or the decision to submit the work for publication.

## Author contributions

Jelena Baranovic, Conceptualization, Data curation, Formal analysis, Investigation, Methodology, Writing—original draft, Writing—review and editing; Andrew JR Plested, Conceptualization, Resources, Software, Formal analysis, Supervision, Funding acquisition, Methodology, Writing—review and editing

## Author ORCIDs

Andrew JR Plested https://orcid.org/0000-0001-6062-0832

## Decision letter and Author response

Decision letter https://doi.org/10.7554/eLife.40548.023
Author response https://doi.org/10.7554/eLife.40548.024

# Additional files

## Supplementary files

• Transparent reporting form
DOI: https://doi.org/10.7554/eLife.40548.021

## Data availability

All data generated or analysed during this study are included in the manuscript. Source data files have been provided for Figure 2, 5 and 6.

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
