## [Decision Letter]

Thank you for submitting your article "Auxiliary subunits keep AMPA receptors compact during activation and desensitization" for consideration by *eLife*. Your article has been reviewed by three peer reviewers, including László Csanády as the Reviewing Editor and Reviewer #1, and the evaluation has been overseen by Richard Aldrich as the Senior Editor. The following individual involved in review of your submission has agreed to reveal their identity: Hiro Furukawa (Reviewer #2).

The reviewers have discussed the reviews with one another and the Reviewing Editor has drafted this decision to help you prepare a revised submission.

Summary:

The study by Baranovic and Plested presents an analysis of the effects of bifunctional cross-linking reagents on homotetrameric AMPA receptors with Cys mutations introduced into the extracellular face of the lower lobe of the ligand binding domain. They find that cross-linking cysteines introduced into the inter-dimer interface (e.g., V666C) causes channel inhibition, whereas cross-linking cysteines introduced into the intra-dimer interface (K493C) causes channel potentiation. The raw data is of very high quality, and some of the experiments are truly heroic, requiring rapid perfusion for patch recordings lasting 20 minutes. The results clearly show auxiliary subunit and state dependence of the inhibitory effect of cross-linking reagents, that will be useful to inform gating models developed using AMPA receptor structures. The effects of Stargazin are dramatic, and nicely anticipated by recent cryo-EM structures from the Gouaux and Sobolevsky labs, but this is not made clear. In summary, the authors have a beautiful set of data that is unique to the field and will be of great value to the community, but in its current form the structural inferences are obscured by muddy presentation. In particular, the reviewers find that the authors do a very poor job of interpreting their data within the library of AMPA receptor structures, and fail to explain whether their data is quantitatively consistent with published structures, or instead suggests that there must be additional conformations the structures of which have yet to be solved. In addition they do not adequately consider what issues might limit the accuracy of the approach used. Therefore, although we do not request additional experiments, a major rewriting will be necessary to make this paper accessible to a broad audience.

Essential revisions:

1) Throughout the manuscript, but especially in the summary and Introduction, the authors use a vague, descriptive structural classification to interpret their results, using the terms 'relaxed' and 'compact' conformations of the 'extracellular layer' of the receptor. Most challenging is the description 'widely splayed, dilated form' of the receptor. It is unclear to the reviewers what they mean by any of these terms, and we suggest that they completely rewrite the paper, using the current library of AMPA receptor cryo-EM structures as a foundation, pointing out where their results either support the structures, or suggest that additional conformations are possible outside those currently determined. Currently they switch repeatedly between attempts to do this and the vague 'relaxed' and 'compact' and 'dilated' descriptions making it very challenging to understand the data in structural terms. It is not even clear if the authors lump conformational changes in the ATD and LBD into their interpretation of 'relaxed' and 'compact' states, of if they mean that the ATD and LBD can independently adopt 'relaxed' and 'compact' conformations, whatever those are.

2) A second issue where substantial improvements could be made would be to explain in the Introduction the anticipated effects expected for any given mutant; the results observed largely reflect trapping of desensitized states, and presumably this was expected based on prior work for mutants at the positions studied. A nice example of how the results of cross-linking experiments were interpreted in a structural framework, comparing anticipated and observed results, is found in the Tajima et al. (2016).

3) When applied to V666C receptors, bis-MTS cross-linkers showed strong effects across all tested lengths between 7 and 18 Å (Figure 2G). The authors argue that "these results demonstrate that desensitized AMPA receptors can 'open up' their LBD layer to ~18 Å at position 666 during desensitization and occupy a spectrum of conformations".

We are uncertain why the fact that an 18 Å cross-linker works should be interpreted to indicate that the separation between the target residues is as large as 18 Å? Can a flexible bifunctional cross-linker not link any two residues that are closer to each other than the extended length of the cross-linker? E.g., wouldn't one expect that a flexible 18 Å cross-linker can cross-link any two sulfhydryl groups that are within 18 Å of each other? In that case, wouldn't two target residues that are positioned at a fixed distance of 7 Å of each other produce a plot just like the one in Figure 2G? The authors should clearly discuss how this issue might limit the accuracy of their approach.

---

## [Author Response]

Essential revisions:1) Throughout the manuscript, but especially in the summary and Introduction, the authors use a vague, descriptive structural classification to interpret their results, using the terms 'relaxed' and 'compact' conformations of the 'extracellular layer' of the receptor. Most challenging is the description 'widely splayed, dilated form' of the receptor. It is unclear to the reviewers what they mean by any of these terms, and we suggest that they completely rewrite the paper, using the current library of AMPA receptor cryo-EM structures as a foundation, pointing out where their results either support the structures, or suggest that additional conformations are possible outside those currently determined. Currently they switch repeatedly between attempts to do this and the vague 'relaxed' and 'compact' and 'dilated' descriptions making it very challenging to understand the data in structural terms. It is not even clear if the authors lump conformational changes in the ATD and LBD into their interpretation of 'relaxed' and 'compact' states, of if they mean that the ATD and LBD can independently adopt 'relaxed' and 'compact' conformations, whatever those are.

Thank you for this important criticism. We have now paid attention to providing more exact descriptions of how we interpret our findings, in particular within the framework of published structural data. We have made adjustments throughout the manuscript.

2) A second issue where substantial improvements could be made would be to explain in the Introduction the anticipated effects expected for any given mutant; the results observed largely reflect trapping of desensitized states, and presumably this was expected based on prior work for mutants at the positions studied. A nice example of how the results of cross-linking experiments were interpreted in a structural framework, comparing anticipated and observed results, is found in the Tajima et al. (2016).

We have now added anticipated cross-linking effects for the mutants, including the LBD intra-dimer K493C mutant, where block of desensitization could be unambiguously predicted, based on published structural and functional data. It is this logic that we followed with the LBD inter-dimer mutants (A665C and V666C): if bis-MTS cross-linkers can constrain dimers to block desensitization, then they should also be able to stabilize active state of the receptor by constraining separation of dimers to distances captured in the structures of the activated receptor (Figure 1C). This aim was not realized, but the distance dependence and state dependence of cross-linker modification is the core of this paper. An important distinction here is between the functional state that favours a given cross-linking event and the effect of the cross-link. In a simple case, these are equivalent – the cross-link occurs in state A and traps the receptor in state A for a long period of time. The reviewers have quite correctly identified that our case with the inter-dimer cross-links in the LBD layer of AMPA receptors is not so simple – the dynamics of the receptor allow it to wriggle between functional states, independent of the state in which a cross-link may be formed.

All this being said, we made extensive changes in the text and we believe we now do a better job of explaining what we found. In so doing, we are sure we improved the manuscript.

3) When applied to V666C receptors, bis-MTS cross-linkers showed strong effects across all tested lengths between 7 and 18 Å (Figure 2G). The authors argue that "these results demonstrate that desensitized AMPA receptors can 'open up' their LBD layer to ~18 Å at position 666 during desensitization and occupy a spectrum of conformations".We are uncertain why the fact that an 18 Å cross-linker works should be interpreted to indicate that the separation between the target residues is as large as 18 Å? Can a flexible bifunctional cross-linker not link any two residues that are closer to each other than the extended length of the cross-linker? E.g., wouldn't one expect that a flexible 18 Å cross-linker can cross-link any two sulfhydryl groups that are within 18 Å of each other? In that case, wouldn't two target residues that are positioned at a fixed distance of 7 Å of each other produce a plot just like the one in Figure 2G? The authors should clearly discuss how this issue might limit the accuracy of their approach.

We thank for reviewers for raising this important point.

First, in addition to flexible cross-linkers, we used the rigid cross-linker bMTSp, which has a length of about 12 Å, and we used it for the experiments in Figure 2G. With regard to the point 3, above, it is hard to rationalize how the fixed 7 Å separation would work so well with bMTSp.

We agree with the reviewers that “one would expect that a flexible 18 Å cross-linker can cross-link (almost) any two sulfhydryl groups that are within 18 Å of each other (there must be limits to the flexibility of the linear bis-MTS)”. But we do not think that “two target residues that are positioned at a fixed distance of 7 Å of each other would produce a plot just like the one in Figure 2G”. The key point is that reacting is not enough, the cross-linker must hinder geometry. A loose cross-linker, less than half extended, cannot offer much hindrance – single reacting MTSEA is without effect in our work. In other words, reacted cross-linkers that are “too long” at all points during the gating cycle should produce little effect, because at no point can they limit the separation of the Cys residues in question.

This line of reasoning is supported by the cross-linking data of the intra-dimer K493C mutant, where a 15 Å-long bis-MTS, M8M, reacts and modifies the current but does not block desensitization because K493C residues do not reach 15 Å-separation during desensitization and their movements are thus not limited in this regard by a slack M8M cross-linker. A similar result was shown in Armstrong et al., 2006. Likewise, the lack of effect of M8M on NMDA receptors carrying Cys mutations in the ATDs (Tajima et al., 2016) was interpreted as the introduced Cys residues being consistently less separated than the length of M8M.

The following explanation was now included in the manuscript to explain our reasoning:

“Because bis-MTS cross-linkers are flexible (except for bMTSp, which is rigid), they can bend and bind to Cys residues whose separation is shorter than the length of the cross-linker. […] Conversely, if the two free Cys residues never separate beyond the length of the cross-linker during gating, then the cross-linker should have little or no effect.”

The following text was added to the Discussion:

“Since most bis-MTS cross-linkers are alkyl chains and hence flexible, they can cross-link pairs of Cys residues closer to each other than the extended length of the cross-linker. […] This is inconsistent with the interpretation of the universal inhibitor and suggests bis-MTS cross-linkers exert their effect mainly by preventing the engineered residues from separating more than the extended length of the cross-linker.”